# Quantitative Propagation of Chaos for SGD in Wide Neural Networks

**Valentin De Bortoli**
University of Oxford
debortoli@stats.ox.ac.uk

**Alain Durmus**
Université Paris-Saclay
alain.durmus@cmla.ens-cachan.fr

**Xavier Fontaine**
Université Paris-Saclay
fontaine@cmla.ens-cachan.fr

**Umut Şimşekli**
LTCI, Télécom Paris, Institut Polytechnique de Paris
umut.simsekli@telecom-paris.fr

## Abstract

In this paper, we investigate the limiting behavior of a continuous-time counterpart of the Stochastic Gradient Descent (SGD) algorithm applied to two-layer overparameterized neural networks, as the number or neurons (*i.e.*, the size of the hidden layer) $N \to +\infty$. Following a probabilistic approach, we show 'propagation of chaos' for the particle system defined by this continuous-time dynamics under different scenarios, indicating that the statistical interaction between the particles asymptotically vanishes. In particular, we establish quantitative convergence with respect to $N$ of any particle to a solution of a mean-field McKean-Vlasov equation in the metric space endowed with the Wasserstein distance. In comparison to previous works on the subject, we consider settings in which the sequence of stepsizes in SGD can potentially depend on the number of neurons and the iterations. We then identify two regimes under which different mean-field limits are obtained, one of them corresponding to an implicitly regularized version of the minimization problem at hand. We perform various experiments on real datasets to validate our theoretical results, assessing the existence of these two regimes on classification problems and illustrating our convergence results.

## 1 Introduction

Due to their ability to tackle very challenging problems, neural networks have been extremely popular and keystones in machine learning [1]. Thanks to their practical success, they have become the de facto tool in many application domains, such as image processing [2] and natural language processing [3]. However, the mathematical understanding of these models and their inherent inference mechanism still remains limited.

Among others, one suprising empirical observation about modern neural networks is that increasing the number of neurons in a network often leads to better classification testing and training errors [4], contradicting the classical statistical learning theory [5]. These experimental results suggest that neural network-based methods exhibit a limiting behavior when the number of neurons is large, *i.e.*, when the neural network is *overparameterized*.

In this paper, we contribute to the recent literature on the theoretical analysis of this phenomenon. To this end, we consider a simple two-layer (*i.e.*, one hidden layer) neural network that is parametrized by $N$ weights $w^{1:N} = \{w^{k,N}\}_{k=1}^N$ and trained to minimize the structural risk $\mathscr{R}^N$ by Stochastic Gradient Descent (SGD) using independent and identically distributed (i.i.d.) samples $(X_i, Y_i)_{i \in \mathbb{N}^\star}$. Even in such a simplified setting, the landscape of $\mathscr{R}^N$ is in many cases arduous to be explored, since $\mathscr{R}^N$ is non-convex and might exhibit many local minima and saddle points [6, 7]; hence making the

minimization of $\mathscr{R}^N$ challenging. However, for large $N$, the analysis of the landscape of $\mathscr{R}^N$ turns out to be much simpler in some situations. For instance [8] has shown that local minima are global minima when the activation function is quadratic as soon as $N$ is larger than twice the size of the original dataset. More generally, relying on approximation or random matrix theory, several works (*e.g.*, [9–19]) establish favorable properties for the landscape of $\mathscr{R}^N$ as $N \to +\infty$, such as absence of saddle points, poor local minima or connected optima. In addition, minimization by SGD in this setting has also proved to be efficient for some models [20, 21].

In this paper we follow an increasingly popular line of research to analyze the behavior of gradient descent-type algorithms (stochastic or deterministic) used for overparameterized models. This approach consists in establishing a 'continuous mean-field limit' for these algorithms as $N \to +\infty$, and has been successively applied in [22–29]. Based on this result, the *qualitative* long-time behavior of SGD applied to overparameterized neural networks can be deduced: these studies all identify an evolution equation on the limiting probability measure which corresponds to a mean-field ordinary differential equation (ODE), *i.e.*, if the initialization is deterministic, then each hidden unit of the network *independently* evolves along the flow of a specific ODE. This implies that, even though the update step is intrinsically stochastic in SGD, the noise completely vanishes in the limit $N \to +\infty$. In this context, two main strategies have been followed to prove convergence of SGD to this mean-field dynamics. The first one is based on gradient flows in Wasserstein spaces [30–32] and the second one is the 'propagation of chaos' phenomenon [33–35], indicating that the statistical interaction between the individual entries of the network asymptotically vanishes. Both approaches are in fact deeply connected, which stems from the duality between probability and partial differential equation theories [36]. We follow in this paper the second approach and establish that propagation of chaos holds for a continuous counterpart of SGD to a solution of a McKean-Vlasov type diffusion [37] as $N \to +\infty$.

The fact that no noise appears in the mean-field limit of SGD obtained in previous work can seem surprising. Aiming to demystify this matter, we study in this paper the case where the stepsize in SGD can depend on the number of neurons. Our main contribution is to identify two mean-field regimes: The first one is the same as the deterministic mean-field limit obtained in the described literature. The second one is a McKean-Vlasov diffusion for which the covariance matrix is non-zero and depends on the properties of the data distribution. To the best of our knowledge, this limiting diffusion has not been reported in the literature and brings interesting insights on the behavior of neural networks in overparameterized settings. Our results suggest that taking large stepsizes in the stochastic optimization procedure corresponds to an *implicit regularization* of the original problem, which can potentially ease the minimization of the structural risk. In addition, in contrast to previous studies, we establish strong quantitative propagation of chaos and we identify the convergence rate of each neuron to its mean-field limit with respect to $N$. Finally we numerically illustrate the existence of these two regimes and the propagation of chaos phenomenon we derive on several classical classification examples on MNIST and CIFAR-10 datasets. In these experiments, the stochastic regime empirically exhibits slightly better generalization properties compared to the deterministic case identified in [22, 23, 28].

## 2 Overparametrized Neural Networks

Consider some feature and label spaces denoted by $\mathsf{X}$ and $\mathsf{Y}$ endowed with $\sigma$-fields $\mathcal{X}$ and $\mathcal{Y}$ respectively. In this paper, we consider a one hidden layer neural network, whose purpose is to classify data from $\mathsf{X}$ with labels in $\mathsf{Y}$. We suppose that the network has $N \in \mathbb{N}^\star$ neurons in the hidden layer whose weights are denoted by $w^{1:N} = \{w^{k,N}\}_{k=1}^N \in (\mathbb{R}^p)^N$. We model the non-linearity by a function $F : \mathbb{R}^p \times \mathsf{X} \to \mathbb{R}$, and consider a loss function $\ell : \mathbb{R} \times \mathsf{Y} \to \mathbb{R}_+$ and a penalty function $V : \mathbb{R}^p \to \mathbb{R}$. Then, the learning problem corresponding to this space of hypothesis consists in minimizing the structural risk

$$\mathscr{R}^N(w^{1:N}) = \int_{\mathsf{X} \times \mathsf{Y}} \ell\left(\frac{1}{N}\sum_{k=1}^N F(w^{k,N}, x), y\right) \mathrm{d}\pi(x,y) + \frac{1}{N}\sum_{k=1}^N V(w^{k,N}),\qquad(1)$$

where $\pi$ is the data distribution on $\mathsf{X} \times \mathsf{Y}$. Note that, in this particular setting, the weights of the second layer are fixed to $(1/N)$. This setting is referred to as "fixed coefficients" in [25, Theorem 1] and is less realistic than the fully-trainable setting. Nevertheless, we believe that this shortcoming can be circumvented upon replacing $F(w^{k,N}, \cdot)$ by $F(u^{k,N}, \cdot)v^{k,N}$ in (1), where $u^{1:N}$ and $v^{1:N}$ are

the weights of the hidden and the second layer respectively. However, this raises new theoretical challenges which are left for future work.

Throughout this paper, we consider the following assumptions.

**A1.** *There exist measurable functions* $\Phi : \mathsf{X} \to [1, +\infty)$ *and* $\Psi : \mathsf{Y} \to [1, +\infty)$ *such that the following conditions hold.*

*(a)* $\ell : \mathbb{R} \times \mathbb{R} \to \mathbb{R}_+$ *is such that for any* $y \in \mathsf{Y}$, $(\tilde{\mathsf{y}} \mapsto \ell(\tilde{\mathsf{y}}, y))$ *is three-times differentiable and for any* $\mathsf{y} \in \mathbb{R}$ *and* $y \in \mathsf{Y}$ *we have*

$$|\partial_1 \ell(0, y)| \leq \Psi(y) , \qquad \left|\partial_1^2 \ell(\mathsf{y}, y)\right| + \left|\partial_1^3 \ell(\mathsf{y}, y)\right| \leq \Psi(y) ,$$

*where for any* $i \in \{1, 2, 3\}$, $\partial_1^i \ell(\mathsf{y}, y)$ *is the* $i$-*th derivative of* $(\tilde{\mathsf{y}} \mapsto \ell(\tilde{\mathsf{y}}, y))$ *at* $\mathsf{y}$.

*(b)* $F : \mathbb{R}^p \times \mathsf{X} \to \mathbb{R}$ *is such that for any* $x \in \mathsf{X}$, $(\tilde{w} \mapsto F(\tilde{w}, x))$ *is three-times differentiable and for any* $w \in \mathbb{R}^p$ *and* $x \in \mathsf{X}$

$$\|F(w, x)\| + \left\|\mathrm{D}_w^1 F(w, x)\right\| + \left\|\mathrm{D}_w^2 F(w, x)\right\| + \left\|\mathrm{D}_w^3 F(w, x)\right\| \leq \Phi(x) ,$$

*where for any* $i \in \{1, 2, 3\}$, $\mathrm{D}_w^i F(w, x)$ *is the* $i$-*th differential of* $(\tilde{w} \mapsto F(\tilde{w}, x))$ *at* $w$.

*(c)* $V \in \mathrm{C}^3(\mathbb{R}^p, \mathbb{R})$ *satisfies* $\sup_{w \in \mathbb{R}^p}\{\|\mathrm{D}^2 V(w)\| + \|\mathrm{D}^3 V(w)\|\} < +\infty$.

*(d) The data distribution* $\pi$ *satisfies* $\int_{\mathsf{X} \times \mathsf{Y}}\{\Phi^{10}(x) + \Psi^4(y)\}\mathrm{d}\pi(x, y) < \infty$ .

Note that **A1**-(d) is immediately satisfied in the case where $\pi$ is compactly supported, $\mathsf{X}$ and $\mathsf{Y}$ are subsets of $\mathbb{R}^d$ and $\mathbb{R}$ respectively and $\Psi$ and $\Phi$ are bounded on the support of $\pi$. **A1**-(a) is satisfied for several losses such as the $\tilde{\mathsf{y}} \mapsto (\tilde{\mathsf{y}} - y)^2$ or the Huber loss $\tilde{\mathsf{y}} \mapsto (1/2)(\tilde{\mathsf{y}} - y)^2$ if $|\tilde{\mathsf{y}} - y| < 1$ and $\tilde{\mathsf{y}} \mapsto |\tilde{\mathsf{y}} - y|$ otherwise. **A1**-(b) is satisfied in the classical case where $F(w, x) = \sigma(\langle w^{(1)}, x\rangle + w^{(2)})$ with $w = (w^{(1)}, w^{(2)})$ and $\sigma$ is the sigmoid function. We believe that the regularity assumptions in **A1** can be relaxed and leave this study for future work. For any $N \in \mathbb{N}^\star$, under **A1**, by the Lebesgue dominated convergence theorem, $\mathscr{R}^N$ given by (1) is well-defined, continuously differentiable with gradient given for any $w^{1:N} \in (\mathbb{R}^p)^N$ by

$$\nabla \mathscr{R}^N(w^{1:N}) = \int_{\mathsf{X} \times \mathsf{Y}} \nabla_w \hat{\mathscr{R}}^N(w^{1:N}, x, y)\mathrm{d}\pi(x, y) ,$$

$$\hat{\mathscr{R}}^N(w^{1:N}, x, y) = \ell\left(\tfrac{1}{N}\textstyle\sum_{k=1}^N F(w^{k,N}, x), y\right) + \tfrac{1}{N}\textstyle\sum_{k=1}^N V(w^{k,N}) ,$$

$$N\nabla_w \hat{\mathscr{R}}^N(w^{1:N}, x, y) = \partial_1 \ell\left(\tfrac{1}{N}\textstyle\sum_{k=1}^N F(w^{k,N}, x), y\right)\nabla_w F^{1:N}(w^{1:N}, x) + \nabla V^{1:N}(w^{1:N}) ,$$

setting $\nabla_w F^{1:N}(w^{1:N}, x) = \{\nabla_w F(w^{k,N}, x)\}_{k=1}^N$, and $\nabla V^{1:N}(w^{1:N}) = \{\nabla V(w^{k,N})\}_{k=1}^N$.

Let $(W_0^k)_{k \in \mathbb{N}^\star}$ be i.i.d. $p$ dimensional random variables with distribution $\mu_0$. Consider the sequence $(W_n^{1:N})_{n \in \mathbb{N}}$ associated with SGD, starting from $W_0^{1:N}$ and defined by the following recursion: for any $n \in \mathbb{N}$ denoting the iteration index

$$W_{n+1}^{1:N} = W_n^{1:N} - \gamma N^\beta(n + \gamma_{\alpha,\beta}(N)^{-1})^{-\alpha}\nabla\hat{\mathscr{R}}^N(W_n^{1:N}, X_n, Y_n) , \tag{2}$$

where $(X_n, Y_n)_{n \in \mathbb{N}}$ is a sequence of i.i.d. input/label samples distributed according to $\pi$, and $(\gamma N^\beta(n + \gamma_{\alpha,\beta}(N)^{-1})^{-\alpha})_{n \in \mathbb{N}}$ as a whole denotes a sequence of stepsizes: here, $\beta \in [0, 1]$, $\alpha \in [0, 1)$, and $\gamma_{\alpha,\beta}(N) = \gamma^{1/(1-\alpha)}N^{(\beta-1)/(1-\alpha)}$. Note that in the constant stepsize setting $\alpha = 0$, the recursion (2) consists in using $\gamma N^\beta$ as a stepsize In addition, it also encompasses the case of decreasing stepsizes (as soon as $\alpha > 0$). The term $\gamma_{\alpha,\beta}(N)^{-1}$ in (2) is a scaling parameter which appears naturally in the corresponding continuous-time dynamics, see (4) below. We stress that contrary to previous approaches such as [22, 23, 28], the stepsize appearing in (2) depends on the number of neurons $N$. Our main contribution is to establish that different mean-field limiting behaviors of a continuous counterpart of SGD arise depending on $\beta$. We emphasize that our results can be extended in a straightforward manner to the case where the sequences of stepsizes is given by $\gamma N^\beta(n + C)^{-\alpha}$ upon replacing $(t + 1)^{-\alpha}$ by $(t + C\gamma_{\alpha,\beta}(N))^{-\alpha}$ in the continuous counterpart of (2).

We will show that the quantity $\gamma_{\alpha,\beta}(N)$ plays the role of a discretization stepsize in the McKean-Vlasov approximation of SGD. The case where $\alpha = 0$ and $\beta = 0$, *i.e.*, the setting considered

by [22, 23, 28], corresponds to choosing the stepsize as $\gamma/N$, which decreases with increasing $N$. In the new setting $\alpha = 0$, $\beta = 1$, this corresponds to take a fixed stepsize $\gamma$. This observation further motivates the scaling and the parameter we introduced in (2).

Before stating our result, we present and give an informal derivation of the continuous particle system dynamics we consider to model (2). We first show that (2) can be rewritten as a recursion corresponding to the discretization of a continuous particle system, *i.e.*, a stochastic differential equation (SDE) with coefficients depending on the empirical measure of the particles. Let us denote by $\mathscr{P}(\mathsf{E})$ the set of probability measures on a measurable space $(\mathsf{E}, \mathcal{E})$. Remark that for each particle dynamics $(W_n^{k,N})_{n \in \mathbb{N}}$ the SGD update (2) is a function of the current position and the empirical measure of the weights. To show this, define the mean-field $h : \mathbb{R}^p \times \mathscr{P}(\mathbb{R}^d) \to \mathbb{R}^p$ and the noise field $\xi : \mathbb{R}^p \times \mathscr{P}(\mathbb{R}^d) \times \mathsf{X} \times \mathsf{Y} \to \mathbb{R}^p$, for any $\mu \in \mathscr{P}(\mathbb{R}^p)$, $w \in \mathbb{R}^p$, $(x, y) \in \mathsf{X} \times \mathsf{Y}$ by

$$h(w, \mu) = -\int_{\mathsf{X} \times \mathsf{Y}} \partial_1 \ell \left( \mu[F(\cdot, x)], y \right) \nabla_w F(w, x) \, \mathrm{d}\pi(x, y) - \nabla V(w) \,,$$

$$\xi(w, \mu, x, y) = -h(w, \mu) - \partial_1 \ell(\mu[F(\cdot, x)], y) \nabla_w F(w, x) - \nabla V(w) \,.$$

Note that with this notation, $h(w^{k,N}, \nu_n^N) = -N\partial_{w^{k,N}} \mathscr{R}^N(w^{1:N})$ and $\xi(w^{k,N}, \nu_n^N, X_n, Y_n) = N\{-\partial_{w^{k,N}} \hat{\mathscr{R}}^N(w^{1:N}, X_n, Y_n) + \partial_{w^{k,N}} \mathscr{R}^N(w^{1:N})\}$, for any $N \in \mathbb{N}$, $k \in \{1, \dots, N\}$ and $n \in \mathbb{N}$, where $\nu^N$ is the empirical measure of the discrete particle system corresponding to SGD defined by $\nu_n^N = N^{-1} \sum_{k=1}^N \delta_{W_n^{k,N}}$. Then, the recursion (2) can be rewritten as follows:

$$W_{n+1}^{k,N} = W_n^{k,N} + \gamma N^{\beta-1}(n + \gamma_{\alpha,\beta}(N)^{-1})^{-\alpha} \left\{ h(W_n^{k,N}, \nu_n^N) + \xi(W_n^{k,N}, \nu_n^N, X_n, Y_n) \right\} \,. \quad (3)$$

We highlight that even though the stepsize in the SGD (2) scales as $\mathcal{O}(N^\beta)$ the stepsize in the associated mean-formulation (3) scales as $\mathcal{O}(N^{\beta-1})$. We now present the continuous model associated with this discrete process. For large $N$ or small $\gamma$ these two processes can be arbitrarily close. For $N \in \mathbb{N}^\star$, consider the particle system diffusion $(\mathbf{W}_t^{1:N})_{t \geq 0} = (\{\mathbf{W}_t^{k,N}\}_{k=1}^N)_{t \geq 0}$ starting from $\mathbf{W}_0^{1:N} = W_0^{1:N}$ defined for any $k \in \{1, \dots, N\}$ by

$$\mathrm{d}\mathbf{W}_t^{k,N} = (t+1)^{-\alpha} \left\{ h(\mathbf{W}_t^{k,N}, \boldsymbol{\nu}_t^N)\mathrm{d}t + \gamma_{\alpha,\beta}(N)^{1/2}\Sigma^{1/2}(\mathbf{W}_t^{k,N}, \boldsymbol{\nu}_t^N)\mathrm{d}\mathbf{B}_t^k \right\} \,, \quad (4)$$

where $\{(\mathbf{B}_t^k)_{t \geq 0} : \quad k \in \mathbb{N}^\star\}$ is a family of independent $p$-dimensional Brownian motions and $\boldsymbol{\nu}_t^N$ is the empirical probability distribution of the particles defined for any $t \geq 0$ by $\boldsymbol{\nu}_t^N = N^{-1} \sum_{k=1}^N \delta_{\mathbf{W}_t^{k,N}}$. In addition in (4), $\Sigma$ is the $p \times p$ matrix given by

$$\Sigma(w, \mu) = \int_{\mathsf{X} \times \mathsf{Y}} \xi(w, \mu, x, y)\xi(w, \mu, x, y)^\top \mathrm{d}\pi(x, y) \,, \qquad \text{for any } w \in \mathbb{R}^p \text{ and } \mu \in \mathscr{P}(\mathbb{R}^p) \,,$$

which is well-defined under **A**1. In the supplementary material we show that under **A**1, (4) admits a unique strong solution. We now give an informal discussion to justify why (4) can be seen as the continuous-time counterpart of (3). For any $N \in \mathbb{N}^\star$, define $(\tilde{\mathbf{W}}_t^{1:N})_{t \geq 0}$ for any $t \geq 0$ by $\tilde{\mathbf{W}}_t^{1:N} = W_{n_t}^{1:N}$ with $n_t = \lfloor t/\gamma_{\alpha,\beta}(N) \rfloor$ and denote $\tilde{\boldsymbol{\nu}}_t^N$ the empirical measure associated with $\tilde{\mathbf{W}}_t^{1:N}$. In this case, by defining the interval $I_{n,\alpha,\beta}^N = [n\gamma_{\alpha,\beta}(N), (n+1)\gamma_{\alpha,\beta}(N)]$ and using (3) and $\gamma_{\alpha,\beta}(N)^{1-\alpha} = \gamma N^{\beta-1}$, we obtain the following approximation for any $n \in \mathbb{N}$

$$\tilde{\mathbf{W}}_{(n+1)\gamma_{\alpha,\beta}(N)}^{k,N} - \tilde{\mathbf{W}}_{n\gamma_{\alpha,\beta}(N)}^{k,N}$$
$$= \gamma N^{\beta-1}(n + \gamma_{\alpha,\beta}(N)^{-1})^{-\alpha} \left\{ h(\tilde{\mathbf{W}}_{n\gamma_{\alpha,\beta}(N)}^{k,N}, \nu_n^N) + \xi(\tilde{\mathbf{W}}_{n\gamma_{\alpha,\beta}(N)}^{k,N}, \nu_n^N, X_n, Y_n) \right\}$$
$$\approx \gamma_{\alpha,\beta}(N)(n\gamma_{\alpha,\beta}(N) + 1)^{-\alpha} \left\{ h(\tilde{\mathbf{W}}_{n\gamma_{\alpha,\beta}(N)}^{k,N}, \tilde{\boldsymbol{\nu}}_{n\gamma_{\alpha,\beta}(N)}^N) + \Sigma^{1/2}(\tilde{\mathbf{W}}_{n\gamma_{\alpha,\beta}(N)}^{k,N}, \tilde{\boldsymbol{\nu}}_{n\gamma_{\alpha,\beta}(N)}^N)G \right\}$$
$$\approx \underbrace{\int_{I_{n,\alpha,\beta}^N} (s+1)^{-\alpha}h(\tilde{\mathbf{W}}_s^{k,N}, \tilde{\boldsymbol{\nu}}_s^N)\mathrm{d}s}_{(A)} + \underbrace{\int_{I_{n,\alpha,\beta}^N} \gamma_{\alpha,\beta}^{1/2}(N)(s+1)^{-\alpha}\Sigma^{1/2}(\tilde{\mathbf{W}}_s^{k,N}, \tilde{\boldsymbol{\nu}}_s^N)\mathrm{d}\mathbf{B}_s^k}_{(B)} \,, \quad (5)$$

where $G$ is a $p$-dimensional Gaussian random variable with zero mean and identity covariance matrix. Note that the second line corresponds to (3) and the last to (4). To obtain such proxy, we first remark that for any $w \in \mathbb{R}^p$ and $\mu \in \mathscr{P}(\mathbb{R}^p)$, $\xi(w, \mu, X_n, Y_n)$ has zero mean and covariance matrix $\Sigma(w, \mu)$

and assume that the noise term is roughly Gaussian. Second, we use that the covariance of $(B)$ in (5) is equal to $\int_{I_{n,\alpha,\beta}^N} \gamma_{\alpha,\beta}(N)(s+1)^{-2\alpha}\Sigma(\tilde{\mathbf{W}}_s^{k,N}, \tilde{\boldsymbol{\nu}}_s^N)\mathrm{d}s$. To obtain the last line, we use some first-order Taylor expansion of this term and $(A)$ as $\gamma_{\alpha,\beta}(N) \to 0$. Then, (5) corresponds to (4) on $I_{n,\alpha,\beta}^N$. As a result, (4) is the continuous counterpart to (3) and $n$ iterations in (3) correspond to the horizon time $n\gamma_{\alpha,\beta}(N)$ in (4). In the next section, we show that a strong quantitative propagation of chaos holds for (4) *i.e.*, we show that for $N \to +\infty$ the particles become indenpendent and have the same distribution associated with a McKean-Vlasov diffusion. The extension of these results to discrete SGD (3) and the rigorous derivation of (5) can be established using strong functional approximations following [38, Proposition 1]. Due to space constraints, we leave it as future work.

Finally, note that until now we only considered the case where the batch size in SGD is equal to one. For a batch size $M \in \mathbb{N}^\star$, this limitation can be lifted replacing $\pi$ and $\hat{\mathscr{R}}^N$ in (2) by $\pi^{\otimes M}$ and

$$\hat{\mathscr{R}}^{N,M}(w^{1:N}, x, y) = \frac{1}{M}\sum_{i=1}^M \ell\left(\frac{1}{N}\sum_{k=1}^N F(w^{k,N}, x_i), y_i\right) ,$$

defined for any $w^{1:N} \in (\mathbb{R}^p)^N$, $x = (x_i)_{i \in \{1,\dots,M\}} \in \mathsf{X}^M$ and $y = (y_i)_{i \in \{1,\dots,M\}} \in \mathsf{Y}^M$. In this case, we obtain that the continuous-time counterpart of (3) is given by (4) upon replacing $\Sigma^{1/2}$ by $\Sigma^{1/2}/M^{1/2}$. This leads to the particle system diffusion $(\mathbf{W}_t^{1:N})_{t\geq 0} = (\{\mathbf{W}_t^{k,N}\}_{k=1}^N)_{t\geq 0}$ starting from $\mathbf{W}_0^{1:N}$ defined for any $k \in \{1,\dots,N\}$ by

$$\mathrm{d}\mathbf{W}_t^{k,N} = (t+1)^{-\alpha}\left\{h(\mathbf{W}_t^{k,N}, \boldsymbol{\nu}_t^N)\mathrm{d}t + (\gamma_{\alpha,\beta}(N)/M)^{1/2}\Sigma^{1/2}(\mathbf{W}_t^{k,N}, \boldsymbol{\nu}_t^N)\mathrm{d}\mathbf{B}_t^k\right\} . \quad (6)$$

In the supplement Section S2, we also present the case of a modified Stochastic Gradient Langevin Dynamics (mSGLD) algorithm [39] which was considered in [23] in the specific case $\beta = 0$. We extend our propagation of chaos results to this setting.

## 3 Mean-Field Approximation and Propagation of Chaos

In this section we identify the mean-field limit of the diffusion (6). More precisely, we show that there exist two regimes depending on how the stepsize scale with the number of hidden units.

Our results are based on the propagation of chaos theory [33–35] and extend the recent works of [22,23, 25,27,28,40]. In what follows, we denote $\mathscr{P}_2(\mathbb{R}^p) = \{\mu \in \mathscr{P}(\mathbb{R}^p) : \int_{\mathbb{R}^p} \|\tilde{w}\|^2 \mathrm{d}\mu(\tilde{w}) < +\infty\}$ and $\mathrm{C}(\mathbb{R}_+, \mathbb{R}^p)$ the set of continuous functions from $\mathbb{R}_+$ to $\mathbb{R}^p$. We also consider the usual metric $\mathrm{m}$ on $\mathrm{C}(\mathbb{R}_+, \mathbb{R}^p)$ defined for any $u_1, u_2 \in \mathrm{C}(\mathbb{R}_+, \mathbb{R}^p)$ by $\mathrm{m}(u_1, u_2) = \sum_{n \in \mathbb{N}^\star} 2^{-n} \|u_1 - u_2\|_{\infty,n}/\{1 + \|u_1 - u_2\|_{\infty,n}\}$, where $\|u_1 - u_2\|_{\infty,n} = \sup_{t\in[0,n]}\|u_1(t) - u_2(t)\|$. It is well-known that $(\mathscr{C}, \mathrm{m}) = (\mathrm{C}(\mathbb{R}_+, \mathbb{R}^p), \mathrm{m})$ is a complete separable space. For any metric space $(\mathsf{F}, \mathrm{m}_\mathsf{F})$, with Borel $\sigma$-field $\mathcal{B}(\mathsf{F})$, we define the extended Wasserstein distance of order 2, denoted $\mathcal{W}_2 : \mathscr{P}(\mathsf{F}) \times \mathscr{P}(\mathsf{F}) \to [0, +\infty]$ for any $\mu_1, \mu_2 \in \mathscr{P}(\mathsf{F})$ by $\mathcal{W}_2^2(\mu_1, \mu_2) = \inf_{\Lambda \in \Gamma(\mu_1,\mu_2)} \int_{\mathsf{F}\times\mathsf{F}} \mathrm{m}_\mathsf{F}^2(v_1, v_2)\mathrm{d}\Lambda(v_1, v_2)$, where $\Gamma(\mu_1, \mu_2)$ is the set of transference plans between $\mu_1$ and $\mu_2$, *i.e.*, $\Lambda \in \Gamma(\mu_1, \mu_2)$ if for any $\mathsf{A} \in \mathcal{B}(\mathsf{F})$, $\Lambda(\mathsf{A} \times \mathsf{F}) = \mu_1(\mathsf{A})$ and $\Lambda(\mathsf{F} \times \mathsf{A}) = \mu_2(\mathsf{A})$.

We start by stating our results in the case where $\beta \in [0, 1)$ for which a deterministic mean-field limit is obtained. Consider the mean-field ODE starting from a random variable $\mathbf{W}_0^\star$ given by

$$\mathrm{d}\mathbf{W}_t^\star = (t+1)^{-\alpha}h(\mathbf{W}_t^\star, \boldsymbol{\lambda}_t^\star)\mathrm{d}t , \qquad \text{with } \boldsymbol{\lambda}_t^\star \text{ the distribution of } \mathbf{W}_t^\star . \quad (7)$$

We show in the supplement that this ODE admits a solution on $\mathbb{R}_+$. This mean-field equation (7) is deterministic conditionally to its initialization.

**Theorem 1.** *Assume* **A**1. *Let* $(\mathbf{W}_0^k)_{k\in\mathbb{N}}$ *be a sequence of i.i.d.* $\mathbb{R}^p$*-valued random variables with distribution* $\mu_0 \in \mathscr{P}_2(\mathbb{R}^p)$ *and set for any* $N \in \mathbb{N}^\star$, $\mathbf{W}_0^{1:N} = \{\mathbf{W}_0^k\}_{k=1}^N$. *Then, for any* $m \in \mathbb{N}^\star$ *and* $T \geq 0$, *there exists* $C_{m,T} \geq 0$ *such that for any* $\alpha \in [0, 1)$, $\beta \in [0, 1)$, $M \in \mathbb{N}^\star$ *and* $N \in \mathbb{N}^\star$ *with* $N \geq m$

$$\mathbb{E}\left[\sup_{t\in[0,T]} \|\mathbf{W}_t^{1:m,N} - \mathbf{W}_t^{1:m,\star}\|^2\right] \leq C_{m,T}\left\{N^{-(1-\beta)/(1-\alpha)}M^{-1} + N^{-1}\right\} ,$$

*with* $(\mathbf{W}_t^{1:m,N}, \mathbf{W}_t^{1:m,\star}) = \{(\mathbf{W}_t^{k,N}, \mathbf{W}_t^{k,\star})\}_{k=1}^m$, $(\mathbf{W}_t^{1:N})$ *the solution of* (6) *starting from* $\mathbf{W}_0^{1:N}$, *and for any* $k \in \mathbb{N}^\star$, $\mathbf{W}_t^{k,\star}$ *the solution of* (7) *starting from* $\mathbf{W}_0^k$.

In Theorem 1, $m$ is a fixed number of particles. Note that $\{(\mathbf{W}_t^{k,\star})_{t\geq 0} : k \in \mathbb{N}^\star\}$ is i.i.d. with distribution $\boldsymbol{\lambda}^\star$ which is the pushfoward measure of $\mu_0$ by the function $(w_0 \mapsto (w_t)_{t\geq 0})$ which from an initial point $w_0$ gives $(w_t)_{t\geq 0} \in \mathscr{C}$ the solution of (7) on $\mathbb{R}_+$. Theorem 1 shows that the dynamics of the particles become deterministic and independent when $N \to +\infty$. The proofs of Theorem 1 and the following result, Theorem 2, are postponed to Section S4.4.

We now consider the case $\beta = 1$ and derive a similar quantitative theorem as Theorem 1 but with a different dynamics than (7). Consider the mean-field SDE starting from variable $\mathbf{W}_0^\star$ given by

$$\mathrm{d}\mathbf{W}_t^\star = (t+1)^{-\alpha}\left\{h(\mathbf{W}_t^\star, \boldsymbol{\lambda}_t^\star)\mathrm{d}t + (\gamma^{1/(1-\alpha)}\Sigma(\mathbf{W}_t^\star, \boldsymbol{\lambda}_t^\star)/M)^{1/2}\mathrm{d}\mathbf{B}_t\right\}, \tag{8}$$

where $\boldsymbol{\lambda}_t^\star$ is the distribution of $\mathbf{W}_t^\star$ and $(\mathbf{B}_t)_{t\geq 0}$ is a $p$ dimensional Brownian motion. Note that taking the limit $\gamma \to 0$ or $M \to +\infty$ in (8) we recover (7). We show in the supplement that this SDE admits a solution on $\mathbb{R}_+$. The following theorem is similar to Theorem 1 in the case $\beta = 1$.

**Theorem 2.** *Let $\beta = 1$. Assume **A**1. Let $(\mathbf{W}_0^k)_{k\in\mathbb{N}}$ be a sequence of $\mathbb{R}^p$-valued random variables with distribution $\mu_0 \in \mathscr{P}_2(\mathbb{R}^p)$ and assume that for any $N \in \mathbb{N}^\star$, $\mathbf{W}_0^{1:N} = \{\mathbf{W}_0^k\}_{k=1}^N$. Then, for any $m \in \mathbb{N}^\star$ and $T \geq 0$, there exists $C_{m,T} \geq 0$ such that for any $\alpha \in [0,1)$, $M \in \mathbb{N}^\star$ and $N \in \mathbb{N}^\star$ with $N \geq m$*

$$\mathbb{E}\left[\sup_{t\in[0,T]}\|\mathbf{W}_t^{1:m,N} - \mathbf{W}_t^{1:m,\star}\|^2\right] \leq C_{m,T}N^{-1},$$

*with $(\mathbf{W}_t^{1:m,N}, \mathbf{W}_t^{1:m,\star}) = \{(\mathbf{W}_t^{k,N}, \mathbf{W}_t^{k,\star})\}_{k=1}^m$, $(\mathbf{W}_t^{1:N})$ the solution of (6) starting from $\mathbf{W}_0^{1:N}$, and for any $k \in \mathbb{N}^\star$, $\mathbf{W}_t^{k,\star}$ the solution of (8) starting from $\mathbf{W}_0^k$ and Brownian motion $(\mathbf{B}_t^k)_{t\geq 0}$.*

The main difference between (7) and (8) is that now this mean-field limit is now longer deterministic up to its initialization but is a SDE driven by a Brownian motion. The stochastic nature of SGD is preserved in this second regime. (7) corresponds to some implicit regularization of (8). In the case where for any $w \in \mathbb{R}^p$ and $\mu \in \mathscr{P}(\mathbb{R}^p)$, $\Sigma(w,\mu) = \sigma^2\,\mathrm{Id}$ with $\sigma > 0$, it can shown that $(\boldsymbol{\lambda}_t^\star)_{t\geq 0}$ is a gradient flow for an entropic-regularized functional. This relation between our approach and the gradient flow perspective is investigated in the supplement Section S6.

Denote for any $N \in \mathbb{N}^\star$ and $m \in \{1,\ldots,N\}$, $\boldsymbol{\lambda}^{1:m,N}$ the distribution on $\mathscr{C}$ of $\{(\mathbf{W}_t^{k,N})_{t\geq 0}\}_{k=1}^m$. Recall that $\{\mathbf{W}_0^{k,N}\}_{k=1}^N$ are $N$ i.i.d. $\mathbb{R}^p$-valued random variables with distribution $\mu_0 \in \mathscr{P}_2(\mathbb{R}^p)$. As an immediate consequence of Theorem 1, Theorem 2 and the definition of $\mathcal{W}_2$ for the distance m on $\mathscr{C}$, we have the following propagation of chaos result.

**Corollary 3.** *Assume **A**1. Then for any $\beta \in [0,1]$, $\alpha \in [0,1)$, $M \in \mathbb{N}^\star$ and $m \in \mathbb{N}$ we have $\lim_{N\to+\infty}\mathcal{W}_2(\boldsymbol{\lambda}^{1:m,N}, (\boldsymbol{\lambda}^\star)^{\otimes m}) = 0$ where $\boldsymbol{\lambda}^\star$ is the distribution of $(\mathbf{W}_t^\star)_{t\geq 0}$ solution of (7) if $\beta \in (0,1]$ and (8) if $\beta = 1$ with $\mathbf{W}_0^\star$ distributed according to $\mu_0$.*

Corollary 3 has two main consequences: when the number of hidden units is large (i) all the units have the same distribution $\boldsymbol{\lambda}^\star$, and (ii) the units are independent. Note also that this corollary is valid for the whole trajectory and not only for a fixed time horizon.

Finally, we derive similar results to Corollary 3 for the sequence of the empirical measures. Let $(\boldsymbol{\nu}^N)_{N\in\mathbb{N}^\star}$ be the sequence of empirical measures associated with (6) and given by $\boldsymbol{\nu}^N = N^{-1}\sum_{k=1}^N \delta_{(\mathbf{W}_t^{k,N})_{t\geq 0}}$. Note that for any $N \in \mathbb{N}$, $\boldsymbol{\nu}^N$ is a random probability measure on $\mathscr{P}(\mathscr{C})$. Denote for any $N \in \mathbb{N}^\star$, $\boldsymbol{\Upsilon}^N$ its distribution which then belongs to $\mathscr{P}(\mathscr{P}(\mathscr{C}))$. Since the convergence with respect to the $\mathcal{W}_2$ distance implies the weak convergence, using Corollary 3 and the Tanaka-Sznitman theorem [33, Proposition 2.2], we get that $(\boldsymbol{\Upsilon}^N)_{N\in\mathbb{N}^\star}$ weakly converges towards $\delta_{\boldsymbol{\lambda}^\star}$. In fact, we prove the following stronger proposition whose proof is postponed to Section S4.3.

**Proposition 4.** *Assume **A**1. Then, for any $\beta \in [0,1)$, $\alpha \in [0,1)$ and $M \in \mathbb{N}^\star$ we have $\lim_{N\to+\infty}\mathcal{W}_2(\boldsymbol{\Upsilon}^N, \delta_{\boldsymbol{\lambda}^\star}) = 0$, where $\boldsymbol{\lambda}^\star$ is the distribution of $(\mathbf{W}_t^\star)_{t\geq 0}$ solution of (7) if $\beta \in (0,1]$ and (8) if $\beta = 1$ with $\mathbf{W}_0^\star$ distributed according to $\mu_0$.*

*Proof of Proposition 4.* We consider only the case $\beta = 1$, the proof for $\beta \in [0,1)$ following the same lines. Let $M \in \mathbb{N}^\star$. We have for any $N \in \mathbb{N}^\star$ using Proposition S1,

$$\mathcal{W}_2(\boldsymbol{\Upsilon}^N, \delta_{\boldsymbol{\lambda}^\star})^2 \leq \mathbb{E}[\mathcal{W}_2(\boldsymbol{\nu}^N, \boldsymbol{\lambda}^\star)^2] \leq N^{-1}\sum_{k=1}^N \mathbb{E}[\mathrm{m}^2((\mathbf{W}_t^{k,N})_{t\geq 0}, (\mathbf{W}_t^{k,\star})_{t\geq 0})]. \tag{9}$$

Let $\varepsilon > 0$ and $n_0 \in \mathbb{N}^\star$ such that $\sum_{n=n_0+1}^{+\infty} 2^{-n} \leq \varepsilon$. Combining (9), Theorem 2 and the Cauchy-Schwarz inequality we get that for any $N \in \mathbb{N}^\star$

$$\mathcal{W}_2(\boldsymbol{\Upsilon}^N, \delta_{\boldsymbol{\lambda}^\star})^2 \leq 2\varepsilon^2 + \frac{2n_0}{N} \sum_{k=1}^{N} \sum_{n=1}^{n_0} \mathbb{E}\left[ \sup_{t \in [0,n]} \|\mathbf{W}_t^{k,N} - \mathbf{W}_t^{k,\star}\|^2 \right] \leq 2\varepsilon^2 + 2n_0 N^{-1} \sum_{n=0}^{n_0} C_{1,n} \; .$$

Therefore, for any $\varepsilon > 0$ there exists $N_0 \in \mathbb{N}^\star$ such that for any $N \geq N_0$, $\mathcal{W}_2(\boldsymbol{\Upsilon}^N, \delta_{\boldsymbol{\lambda}^\star}) \leq \varepsilon$. $\qquad \square$

**Relation to existing results.** To the authors knowledge, only the case $\beta = 0$ has been considered in the current literature. More precisely, Theorem 1 is a functional and quantitative extension of the results established in [22–24, 28, 40]. First, in [22, Theorem 1.6], it is shown that $(\boldsymbol{\lambda}^{1:m,N})_{N \in \mathbb{N}^\star, N \geq \ell}$ weakly converges towards $(\boldsymbol{\lambda}^\star)^{\otimes m}$. [23, Theorem 3] shows quantitative weak convergence of SGD to (7) with high probability in the case $V = 0$ and the quadratic loss $\ell(\mathrm{y}_1, \mathrm{y}_2) = (\mathrm{y}_1 - \mathrm{y}_2)^2$. [40, Theorem 1.5] establishes a central limit theorem for $(\boldsymbol{\nu}^N)_{N \in \mathbb{N}^\star}$ with rate $N^{-1/2}$ which is in accordance with the convergence rate identified in Theorem 1. Note that the identified convergence rates in Theorem 1 and Theorem 2 do not depend on the decreasing rate of the stepsizes given by $\alpha$. We expect an interplay between $\alpha$ and $\beta$ to appear in a the rates of a CLT for $(\boldsymbol{\nu}^N)_{N \in \mathbb{N}^\star}$ in our setting. The rigorous derivation of such results is left for future work. Finally, [28, Theorem 2.6] and [24, Proposition 3.2] imply the convergence of $\boldsymbol{\nu}^N$ almost surely under the setting $\Sigma = 0$ in (6) which corresponds to the continuous gradient flow dynamics associated with $\hat{\mathscr{R}}^{N,M}$. We conclude this part by mentioning that similar results are derived for mSGLD in the supplement Section S2 which extend the ones obtained in [25].

Having established the convergence of (6) to (8), we are interested in the long-time behaviour of $(\mathbf{W}_t^\star)_{t \geq 0}$ in the case $\alpha = 0$. To address this problem, the first step is to show that this SDE admits at least one stationary distribution, *i.e.* a probability measure $\mu^\star$ such that if $\mathbf{W}_0^\star$ has distribution $\mu^\star$, then for any $t \geq 0$, $\mathbf{W}_t^\star$ has distribution $\mu^\star$. If $V$ is strongly convex, we are able to answer positively to this question in the case $p = 1$. The proof of this result is postponed to Section S5.

**Proposition 5.** *Assume* A1*, $\alpha = 0$ and $p = 1$. In addition, assume that there exist $\eta, \bar{\sigma} > 0$ such that for any $w \in \mathbb{R}$ and $\mu \in \mathscr{P}(\mathbb{R})$, $\Sigma(w, \mu) \geq \bar{\sigma}^2$ and $V$ is $\eta$-strongly convex. Let $H : \mathscr{P}_2(\mathbb{R}) \to \mathscr{P}_2(\mathbb{R})$ defined for any $\mu \in \mathscr{P}_2(\mathbb{R})$ and $w \in \mathbb{R}$ by*

$$(\mathrm{d}H(\mu)/\mathrm{dLeb})(w) \propto \bar{\Sigma}^{-1}(w, \mu) \exp\left[ -2 \int_0^w h(\tilde{w}, \mu)/\bar{\Sigma}(\tilde{w}, \mu) \mathrm{d}\tilde{w} \right] \; ,$$

*where $\bar{\Sigma}(w, \mu) = \gamma^{1/(1-\alpha)} \Sigma(w, \mu)/M$ and* Leb *is the Lebesgue measure on $\mathbb{R}$. Then* $\mathsf{S} = \{\mu \in \mathscr{P}_2(\mathbb{R}) : H(\mu) = \mu\} \neq \emptyset$ *and for any $\mu \in \mathsf{S}$, $\mu$ is invariant for (8).*

## 4 Experiments

We now empirically illustrate the results derived in the previous section. More precisely, we focus on the classification task for two datasets: MNIST [41] and CIFAR-10 [42]. In all of our experiments we consider a fully-connected neural network with one hidden layer and ReLU activation function. We consider the cross-entropy loss in order to train the neural network using SGD as described in Section 2. All along this section we fix a time horizon $T \geq 0$ and sample $W_{n_T}^{1:N}$ defined by (3) with $n_T = \lfloor T/\gamma_{\alpha,\beta}(N) \rfloor$ for $\gamma_{\alpha,\beta}(N) = \gamma^{1/(1-\alpha)} N^{(\beta-1)/(1-\alpha)}$ and taking a batch of size $M \in \mathbb{N}^\star$. We aim at illustrating the results of Section 3 taking $N \to +\infty$ and different sets of values for the parameters $\alpha, \beta, M, \gamma$ in (6). Indeed, recall that as observed in (5), $W_{n_T}^{1:N}$ is an approximation of $\mathbf{W}_T^{1:N}$. See Section S7 for a detailed description of our experimental setting. If not specified, we set $\alpha = 0$, $M = 100$, $T = 100$, $\gamma = 1$.

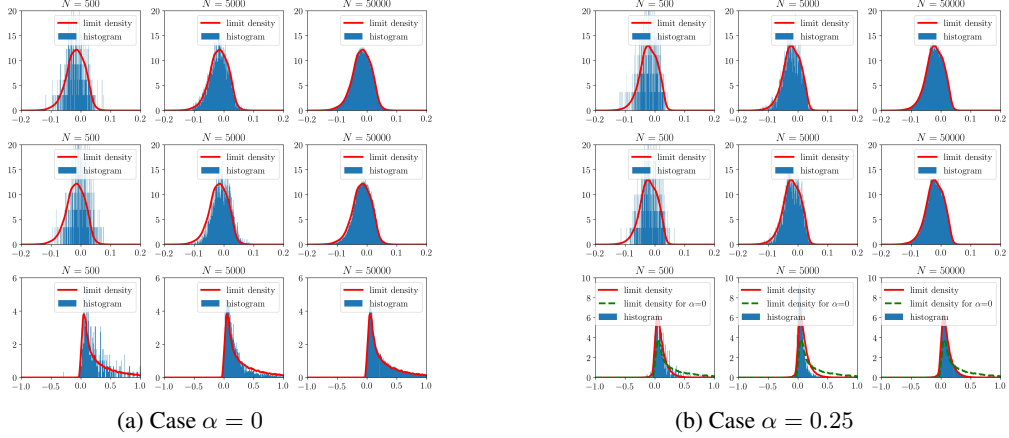

(a) Case $\alpha = 0$            (b) Case $\alpha = 0.25$

Figure 1: Convergence of the empirical distribution of the weights as $N \to +\infty$.

**Convergence of the empirical measure.** First we assess the convergence of the empirical distribution of the weights of the hidden layer to a limit distribution when $N \to +\infty$. We focus on the MNIST classification task. Note that in this case $p = 28 \times 28 = 784$. In Figure 1, we observe the behavior of the histograms of the weights $W_{n_T}^{1:N}$ of the hidden layer along the coordinate $(1,1)$ as $N \to +\infty$. We experimentally observe the existence of two different regimes, one for $\beta < 1$ and the other one for $\beta = 1$. In Figure 1, the first line corresponds to the evolution of the histogram in the case where $\beta = 0.5$. The second and the third lines correspond to the same experiment with $\beta = 0.75$ and $\beta = 1$, respectively. Note that in both cases the histograms converge to a limit. This limit histogram exhibits two regimes depending if $\beta < 1$ or $\beta = 1$.

**Existence of two regimes.** Now we assess the stochastic nature of the second regime we obtain in the case $\beta = 1$ in contrast to the regime for $\beta < 1$ which is deterministic. In order to highlight this situation, all the weights of the neural network are initialized with a fixed value, *i.e.*, for any $N \in \mathbb{N}^\star$ and $k \in \{1, \dots, N\}$, $W_0^{k,N} = w_0 \in \mathbb{R}^p$. Then, the neural network is trained on the MNIST dataset for $N = 10^6$ and $\beta = 0.75$ or $\beta = 1$. Figure 2 represents 7 samples of the first component of $W_{n_T}^{1:N}$ obtained

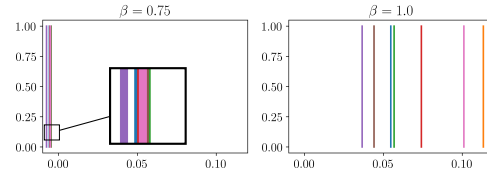

Figure 2: Deterministic versus stochastic behavior depending on the value $\beta$.

with independent runs of SGD. We can observe that for $\beta = 0.75$ all the samples converge to the same value which agrees with (7) while in the case where $\beta = 1$ they exhibit different values, which is in accordance with (8).

**From stochastic to deterministic.** We illustrate that when $\gamma \to 0$ the dynamics identified in (8) tends to the one identified in (7). We fix $\beta = 1$ and $N = 10000$ and focus on the MNIST classification task. In Figure 3 we show the histogram of the weights $W_{n_T}^{1:N}$ along the coordinate $(1,1)$ for different values of $\gamma$. As expected, see (8) and the follow-

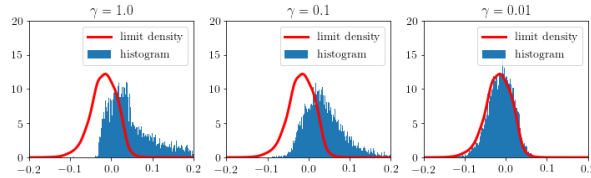

Figure 3: Convergence to the deterministic regime as $\gamma \to 0$.

ing remark, when $\gamma \to 0$ we recover the limit histogram with $\beta < 1$. In Figure S5 we also study the convergence of the empirical measure when $M \to +\infty$ in the case where $\beta < 1$.

**Long-time behavior.** Finally, we illustrate the interest of taking $\beta = 1$ in our setting by considering the more challenging classification task on the CIFAR-10 dataset. We consider the following set of parameters $\alpha = 0$, $M = 100$, $T = 10000$, $\gamma = 0.1$. We emphasize that this experiment aims at comparing the performance of the setting $\beta < 1$ and the one with $\beta = 1$ and that we are not trying to

reach state-of-the-art results. In Table 1 we present training and test accuracies for the classification task at hand. To build the classification estimator we average the weights along their trajectory, *i.e.*, we perform averaging and consider the average estimator $\bar{W}_{n_T}^{1:N} = (n_T - n_0 + 1)^{-1} \sum_{n=n_0}^{n_T} W_n^{1:N}$, where $n_0 = 1000$. Using $\beta = 1$ roughly increases the test accuracy by $1\%$, while the training accuracy is not $100\%$. This empirically seems to demonstrate that using a smaller value of $\beta$ tends to overfit the data, whereas using $\beta = 1$ has a regularizing effect.

Table 1: Training and Test accuracies for different settings on the CIFAR-10 dataset, with $\alpha = 0$, $M = 100$ and $\gamma = 0.1$ for $T = 10000$

| Values of $N$ and $\beta$ | $N = 5000$ $\beta = 0.75$ | $N = 5000$ $\beta = 1.0$ | $N = 10000$ $\beta = 0.75$ | $N = 10000$ $\beta = 1.0$ | $N = 50000$ $\beta = 0.75$ | $N = 50000$ $\beta = 1.0$ |
|---|---|---|---|---|---|---|
| Train acc. | **100%** | 97.2% | **100%** | 97.2% | **100%** | 99% |
| Test acc. | 55.5% | **56.5%** | 56.0% | **56.5%** | 56.7% | **57.7%** |

## 5   Conclusion

We show in this paper that taking a stepsize in SGD depending on the number of hidden units leads to particle systems with two possible mean-field behaviours. The first was already identified in [22, 23, 28] and corresponds to a deterministic mean-field ODE. The second is new and corresponds to a McKean-Vlasov diffusion. Our numerical experiments on two real datasets support our findings. In a future work, we intend to follow the same approach for deep neural networks, *i.e.*, with a growing number of hidden layers.

## Acknowledgments and Disclosure of Funding

V. De Bortoli was partially supported by EPSRC grant EP/R034710/1. X. Fontaine was supported by grants from Région Ile-de-France. The contribution of U. Şimşekli to this work is partly supported by the French National Research Agency (ANR) as a part of the FBIMATRIX (ANR-16-CE23-0014) project.

## Broader Impact

Today neural networks can be seen by the non-expert public as black boxes and the lack of understanding of the underlying mechanisms in deep learning can frighten some decision makers to use them due to the risk of unexplained failures.

A theoretical understanding of the behavior of usual training algorithms such as SGD when applied to neural networks is indeed still largely missing. Such a theory would benefit the experimentators who could use these results to design new algorithms and improve existing ones. For instance, deriving properties on the limit points of SGD would give more insight on the estimators constructed by training neural networks. In particular, we could characterize their bias and dependency with respect to the training data and quantify their generalization properties.

Analyzing the long-time behavior of classical training algorithms such as SGD is crucial to pinpoint what are the essential properties of neural networks.

We show that in the simplistic case of a two-layer neural network SGD admits a limiting dynamics when the number of hidden units $N \to +\infty$. We identify two regimes depending on an internal parameter $\beta \in [0, 1]$. If $\beta \in [0, 1)$ we recover known deterministic mean-field dynamics and in the case $\beta = 1$ we obtain a stochastic mean-field dynamics. Therefore, we claim that, at least in the case of two-layer neural networks, the long-time behavior of SGD can be analyzed using these mean-field approximations. It constitutes a step towards a deeper understanding of their properties. We focus on the simplistic case of a two-layer neural network.

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
