[Supplementary Material]

# Quantitative Propagation of Chaos for SGD in Wide Neural Networks

## SUPPLEMENTARY DOCUMENT

**Valentin De Bortoli**
University of Oxford
debortoli@stats.ox.ac.uk

**Xavier Fontaine**
Université Paris-Saclay
fontaine@cmla.ens-cachan.fr

**Alain Durmus**
Université Paris-Saclay
alain.durmus@cmla.ens-cachan.fr

**Umut Şimşekli**
Telecom Paris
umut.simsekli@telecom-paris.fr

## Contents

# S1 Preliminaries

## S1.1 Notation

Let $(\mathsf{E}, d_E)$ and $(\mathsf{F}, d_F)$ be two metric spaces. $\mathrm{C}(\mathsf{E}, \mathsf{F})$ stands for the set of continuous $\mathsf{F}$-valued functions. If $\mathsf{F} = \mathbb{R}$, then we simply note $\mathrm{C}(\mathsf{E})$.

We say that $f : \mathsf{E} \to \mathbb{R}^p$ is $L$-Lipschitz if there exists $L \geq 0$ such that for any $x, y \in \mathsf{E}$, $\|f(x) - f(y)\| \leq L d_E(x, y)$. Let $\mathrm{C}_{\mathrm{b}}(\mathsf{E}, \mathbb{R}^p)$ (respectively $\mathrm{C}_{\mathrm{c}}(\mathsf{E}, \mathbb{R}^p)$) be the set of bounded continuous functions from $\mathsf{E}$ to $\mathbb{R}^p$ (respectively the set of compactly supported functions from $\mathsf{E}$ to $\mathbb{R}^p$). If $p = 1$, we simply note $\mathrm{C}_{\mathrm{b}}(\mathsf{E})$ (respectively $\mathrm{C}_{\mathrm{c}}(\mathsf{E})$).

For $\mathsf{U}$ an open set of $\mathbb{R}^d$, $n \in \mathbb{N}^\star$ and define $\mathrm{C}^n(\mathsf{U}, \mathbb{R}^p)$ the set of the $n$-differentiable $\mathbb{R}^p$-valued functions over $\mathsf{U}$. If $p = 1$ then we simply note $\mathrm{C}^n(\mathsf{U})$. Let $f \in \mathrm{C}^1(\mathsf{U})$ we denote by $\nabla f$ its gradient. More generally, if $f \in \mathrm{C}^n(\mathsf{U}, \mathbb{R}^p)$ with $n, p \in \mathbb{N}^\star$, we denote by $\mathrm{D}^k f(x)$ the $k$-th differential of $f$. We also denote for any $i \in \{1, \ldots, d\}$ and $\ell \in \{1, \ldots, k\}$, $\partial_i^\ell f$ the $i$-th partial derivative of $f$ of order $\ell$. If $f \in \mathrm{C}^2(\mathbb{R}^d, \mathbb{R})$, we denote by $\Delta f$ its Laplacian. $\mathrm{C}_{\mathrm{c}}^n(\mathsf{U}, \mathbb{R}^p)$ is the subset of $\mathrm{C}^n(\mathsf{U}, \mathbb{R}^p)$ such that for any $f \in \mathrm{C}_{\mathrm{c}}^n(\mathsf{U}, \mathbb{R}^p)$ and $\ell \in \{0, \ldots, n\}$, $\mathrm{D}^\ell f$ has compact support.

Consider $(\mathsf{F}, d)$ a metric space. Let $\mathscr{P}(\mathsf{F})$ be the space of probability measures over $\mathsf{F}$ equipped with its Borel $\sigma$-field $\mathcal{B}(\mathsf{F})$. For any $\mu \in \mathscr{P}(\mathsf{F})$ and $f : \mathsf{F} \to \mathbb{R}$, we say that $f$ is $\mu$-integrable if $\int_{\mathsf{F}} |f(x)| \mathrm{d}\mu(x) < +\infty$. In this case, we set $\mu[f] = \int_{\mathsf{F}} f(x) \mathrm{d}\mu(x)$. Let $\mu_0 \in \mathscr{P}(\mathsf{F})$. For any $r \geq 1$, define $\mathscr{P}_r(\mathsf{F}) = \{\mu \in \mathscr{P}(\mathsf{F}) : \int_{\mathbb{R}^p} d(\mu_0, \mu)^r \mathrm{d}\mu(x) < +\infty\}$. If not specified, we consider a filtered probability space $(\Omega, \mathcal{F}, \mathbb{P}, (\mathcal{F}_t)_{t \geq 0})$ satisfying the usual conditions and any random variables is defined on this probability space. Let $f : (\mathsf{E}, \mathcal{E}) \to (\mathsf{G}, \mathcal{G})$ be a measurable function. Then for any measure $\mu$ on $\mathcal{E}$ we define its pushforward measure by $f$, $f_\# \mu$, for any $\mathsf{A} \in \mathcal{G}$ by $f_\# \mu(\mathsf{A}) = \mu(f^{-1}(\mathsf{A}))$.

The set of $m \times n$ real matrices is denoted by $\mathbb{R}^{m \times n}$. The set of symmetric real matrices of size $p$ is denoted $\mathbb{S}_p(\mathbb{R})$.

## S1.2 Wasserstein distances

Let $(\mathsf{F}, d)$ be a metric space. Let $\mu_1, \mu_2 \in \mathscr{P}(\mathsf{F})$, where $\mathsf{F}$ is equipped with its Borel $\sigma$-field $\mathcal{B}(\mathsf{F})$. A probability measure $\zeta$ over $\mathcal{B}(\mathsf{F})^{\otimes 2}$ is said to be a transference plan between $\mu_1$ and $\mu_2$ if for any $\mathsf{A} \in \mathcal{B}(\mathsf{F})$, $\zeta(\mathsf{A} \times \mathsf{F}) = \mu_1(\mathsf{A})$ and $\zeta(\mathsf{F} \times \mathsf{A}) = \mu_2(\mathsf{A})$. We denote by $\Lambda(\mu_1, \mu_2)$ the set of all transference plans between $\mu_1$ and $\mu_2$. If $\mu_1, \mu_2 \in \mathscr{P}_r(\mathbb{R}^p)$, we define the Wasserstein distance $\mathscr{W}_r(\mu_1, \mu_2)$ of order $r$ between $\mu_1$ and $\mu_2$ by

$$\mathscr{W}_r^r(\mu_1, \mu_2) = \inf_{\zeta \in \Lambda(\mu_1, \mu_2)} \left\{ \int_{\mathsf{F} \times \mathsf{F}} d(x, y)^r \mathrm{d}\zeta(x, y) \right\} . \tag{S1}$$

Note that $\mathscr{W}_r$ is a distance on $\mathscr{P}_r(\mathsf{F})$ by [1, Theorem 6.18]. In addition $(\mathscr{P}_r(\mathbb{R}^p), \mathscr{W}_r)$ is a complete separable metric space. For any $\mu_1, \mu_2 \in \mathscr{P}_p(\mathsf{F})$ we say that a couple of random variables $(X, Y)$ is an optimal coupling of $(\mu_1, \mu_2)$ for $\mathscr{W}_p$ if it has distribution $\xi$ where $\xi$ is an optimal transference plan between $\mu_1$ and $\mu_2$.

For any $T \geq 0$, the space $\mathscr{C}_{2,T}^p = \mathrm{C}([0, T], \mathscr{P}_2(\mathbb{R}^p))$ is a complete separable metric space [2, Theorem 4.19] with the metric $\mathscr{W}_{2,T}$ given for any $(\nu_t)_{t \in [0,T]}$ and $(\mu_t)_{t \in [0,T]}$ by

$$\mathscr{W}_{2,T}((\nu_t)_{t \in [0,T]}, (\mu_t)_{t \in [0,T]}) = \sup_{t \in [0,T]} \mathscr{W}_2(\nu_t, \mu_t) .$$

In the case where the measures we consider can be written as sums of Dirac we have the following proposition.

**Proposition S1.** *Let $r \geq 1$, $N \in \mathbb{N}^\star$, $\{\alpha_k\}_{k=1}^N \in [0, 1]^N$ with $\sum_{k=1}^N \alpha_k = 1$, $\{\mu_{k,a}\}_{k=1}^N \in \mathscr{P}(\mathsf{F})^N$ and $\{\mu_{k,b}\}_{k=1}^N \in \mathscr{P}(\mathsf{F})^N$. Then, setting $\nu_i = \sum_{k=1}^N \alpha_k \mu_{k,i}$ with $i \in \{a, b\}$, we have*

$$\mathscr{W}_r^r(\nu_a, \nu_b) w \leq \sum_{k=1}^N \mathscr{W}_r^r(\mu_{k,a}, \mu_{k,b}) .$$

*Proof.* Consider $\zeta = \sum_{k=1}^{N} \alpha_k \zeta_k \in \Lambda(\nu_a, \nu_b)$ with $\zeta_k$ the optimal transference plan between $\mu_{k,a}$ and $\mu_{k,b}$. Then, we have

$$\mathcal{W}_r^r(\nu_a, \nu_b) \leq \int_{\mathbb{R}^p \times \mathbb{R}^p} d(x,y)^r \mathrm{d}\zeta(x,y) \leq N^{-1} \sum_{k=1}^{N} \mathcal{W}_r^r(\mu_{k,a}, \mu_{k,b}) \, .$$

$\square$

As a special case of Proposition S1, we obtain that for any $r \geq 1$, $\{w_{k,a}\}_{k=1}^{N} \in \mathsf{F}^N$ and $\{w_{k,a}\}_{k=1}^{N} \in \mathsf{F}^N$,

$$\mathcal{W}_r(N^{-1} \sum_{k=1}^{N} \delta_{w_{k,a}}, N^{-1} \sum_{k=1}^{N} \delta_{w_{k,b}}) \leq N^{-1} \sum_{k=1}^{N} d(w_{k,a}, w_{k,b})^r \, .$$

As another special case of Proposition S1, we obtain that for any $\mu \in \mathscr{P}_r(\mathsf{F})$ and $\{w_k\}_{k=1}^{N} \in \mathsf{F}^N$

$$\mathcal{W}_r(N^{-1} \sum_{k=1}^{N} \delta_{w_k}, N^{-1}, \mu) \leq N^{-1} \sum_{k=1}^{N} \mathcal{W}_r(w_k, \mu)^r \, .$$

## S2 A mean-field modification of Stochastic Gradient Langevin Dynamics

### S2.1 Presentation of the modified SGLD and its continuous counterpart

We start by introducing a modified Stochastic Gradient Langevin Dynamics (mSGLD) [3]. In the mean-field regime, this setting was studied in the case $\beta = 0$ in [4]. We recall that the mean-field $h : \mathbb{R}^p \times \mathscr{P}(\mathbb{R}^d) \to \mathbb{R}^p$ and $\xi : \mathbb{R}^p \times \mathscr{P}(\mathbb{R}^d) \times \mathsf{X} \times \mathsf{Y} \to \mathbb{R}^p$ are given for any $\mu \in \mathscr{P}(\mathbb{R}^p)$, $w \in \mathbb{R}^p$, $(x,y) \in \mathsf{X} \times \mathsf{Y}$ by

$$h(w, \mu) = -\int_{\mathsf{X} \times \mathsf{Y}} \partial_1 \ell\left(\mu[F(\cdot, x)], y\right) \nabla_w F(w, x) \, \mathrm{d}\pi(x,y) - \nabla V(w) \, ,$$

$$\xi(w, \mu, x, y) = -h(w, \mu) - \partial_1 \ell(\mu[F(\cdot, x)], y) \nabla_w F(w, x) - \nabla V(w) \, .$$

Let $(W_0^k)_{k \in \mathbb{N}^\star}$ be i.i.d. $p$ dimensional random variables with distribution $\mu_0$ and $\{Z_k^n : k, n \in \mathbb{N}^\star\}$ be i.i.d. $p$ dimensional independent Gaussian random variables with zero mean and identity covariance matrix. Consider the sequence $(W_n^{1:N})_{n \in \mathbb{N}}$ associated with mSGLD starting from $W_0^{1:N}$ and defined by the following recursion: for any $n \in \mathbb{N}$, $k \in \{1, \ldots, N\}$,

$$W_{n+1}^{k,N} = W_n^{k,N} + \gamma N^{\beta-1}(n + \gamma_{\alpha,\beta}(N)^{-1})^{-\alpha} \left\{ h(W_n^{k,N}, \nu_n^N) + \xi(W_n^{k,N}, \nu_n^N, X_n, Y_n) \right\}$$
$$+ \left[ 2\eta\gamma N^{\beta-1}(n + \gamma_{\alpha,\beta}(N)^{-1})^{-\alpha} \right]^{1/2} Z_{k,n} \, , \quad \text{(S2)}$$

where $\eta \geq 0$, $\beta \in [0,1]$, $\alpha \in [0,1)$, $\gamma > 0$, $(X_n, Y_n)_{n \in \mathbb{N}}$ is a sequence of i.i.d. input/label samples distributed according to $\pi$ and $\gamma_{\alpha,\beta}(N) = \gamma^{1/(1-\alpha)} N^{(\beta-1)/(1-\alpha)}$. Note that in the case $\eta = 0$, we obtain (3). In addition, (S2) does not exactly correspond to the usual implementation of SGLD as introduced in [3]. Indeed, to recover this algorithm, we should replace $[2\eta\gamma N^{\beta-1}(n + \gamma_{\alpha,\beta}(N)^{-1})^{-\alpha}]^{1/2} Z_{k,n}$ by $[2\eta\gamma N^{\beta}(n + \gamma_{\alpha,\beta}(N)^{-1})^{-\alpha}]^{1/2} Z_{k,n}$ in (S2). The scheme presented in (S2) amounts to consider a temperature which scales as $\gamma N^{\beta-1}$ with the number of particles. As emphasized before, this scheme was also considered in [4].

We now present the continuous model associated with this discrete process in the limit $\gamma \to 0$ or $N \to +\infty$. For $N \in \mathbb{N}^\star$, consider the particle system diffusion $(\mathbf{W}_t^{1:N})_{t \geq 0} = (\{\mathbf{W}_t^{k,N}\}_{k=1}^{N})_{t \geq 0}$ starting from $\mathbf{W}_0^{1:N}$ defined for any $k \in \{1, \ldots, N\}$ by

$$\mathrm{d}\mathbf{W}_t^{k,N} = (t+1)^{-\alpha} \left\{ h(\mathbf{W}_t^{k,N}, \boldsymbol{\nu}_t^N) \mathrm{d}t + \gamma_{\alpha,\beta}(N)^{1/2} \Sigma^{1/2}(\mathbf{W}_t^{k,N}, \boldsymbol{\nu}_t^N) \mathrm{d}\mathbf{B}_t^k + \sqrt{2\eta}\mathrm{d}\tilde{\mathbf{B}}_t^k \right\} \, , \quad \text{(S3)}$$

where $\{(\mathbf{B}_t^k)_{t \geq 0} : k \in \mathbb{N}^\star\}$ and $\{(\tilde{\mathbf{B}}_t^k)_{t \geq 0} : k \in \mathbb{N}^\star\}$ are two independent families of independent $p$ dimensional Brownian motions and $\boldsymbol{\nu}_t^N$ is the empirical probability distribution of the particles defined for any $t \geq 0$ by $\boldsymbol{\nu}_t^N = N^{-1} \sum_{k=1}^{N} \delta_{\mathbf{W}_t^{k,N}}$. Similarly to Section 2, (S3) is the continuous

counterpart of (S2). Let $M \in \mathbb{N}^\star$. Similarly to (6), we consider the following particle system diffusion $(\mathbf{W}_t^{1:N})_{t \geq 0} = (\{\mathbf{W}_t^{k,N}\}_{k=1}^N)_{t \geq 0}$ starting from $\mathbf{W}_0^{1:N}$ defined for any $k \in \{1, \dots, N\}$ by

$$\mathrm{d}\mathbf{W}_t^{k,N} = (t+1)^{-\alpha} \left\{ h(\mathbf{W}_t^{k,N}, \boldsymbol{\nu}_t^N)\mathrm{d}t + (\gamma_{\alpha,\beta}(N)/M)^{1/2}\Sigma^{1/2}(\mathbf{W}_t^{k,N}, \boldsymbol{\nu}_t^N)\mathrm{d}\mathbf{B}_t^k + \sqrt{2\eta}\mathrm{d}\tilde{\mathbf{B}}_t^k \right\} . \tag{S4}$$

### S2.2 Mean field approximation and propagation of chaos for mSGLD

The following theorems are the extensions of Theorem 1 and Theorem 2 to (S3) for any $\eta \geq 0$. Note that in the case $\eta = 0$, Theorem S2 boils down to Theorem 1 and Theorem S3 to Theorem 2.

We start by stating our results in the case $\beta \in [0, 1)$. Consider the mean-field SDE starting from a random variable $\mathbf{W}_0^\star$ given by

$$\mathrm{d}\mathbf{W}_t^\star = (t+1)^{-\alpha} \left\{ h(\mathbf{W}_t^\star, \boldsymbol{\lambda}_t^\star)\mathrm{d}t + \sqrt{2\eta}\tilde{\mathbf{B}}_t \right\} , \qquad \text{with } \boldsymbol{\lambda}_t^\star \text{ the distribution of } \mathbf{W}_t^\star . \tag{S5}$$

**Theorem S2.** *Assume* **A**1. *Let* $(\mathbf{W}_0^k)_{k \in \mathbb{N}}$ *be a sequence of i.i.d.* $\mathbb{R}^p$-*valued random variables with distribution* $\mu_0 \in \mathscr{P}_2(\mathbb{R}^p)$ *and set for any* $N \in \mathbb{N}^\star$, $\mathbf{W}_0^{1:N} = \{\mathbf{W}_0^k\}_{k=1}^N$. *Then, for any* $m \in \mathbb{N}^\star$ *and* $T \geq 0$, *there exists* $C_{m,T} \geq 0$ *such that for any* $\alpha \in [0, 1)$, $\beta \in [0, 1)$, $M \in \mathbb{N}^\star$ *and* $N \in \mathbb{N}^\star$

$$\mathbb{E}\left[\sup_{t \in [0,T]} \|\mathbf{W}_t^{1:m,N} - \mathbf{W}_t^{1:m,\star}\|^2\right] \leq C_{m,T} \left\{ N^{-(1-\beta)/(1-\alpha)}M^{-1} + N^{-1} \right\} ,$$

*with* $(\mathbf{W}_t^{1:m,N}, \mathbf{W}_t^{1:m,\star}) = \{(\mathbf{W}_t^{k,N}, \mathbf{W}_t^{k,\star})\}_{k=1}^m$, $(\mathbf{W}_t^{1:N})$ *is the solution of* (S4) *starting from* $\mathbf{W}_0^{1:N}$, *and for any* $k \in \{1, \dots, N\}$, $\mathbf{W}_t^{k,\star}$ *is the solution of* (S5) *starting from* $\mathbf{W}_0^k$ *and Brownian motion* $(\tilde{\mathbf{B}}_t^k)_{t \geq 0}$.

*Proof.* The proof is postponed to Section S4.4 $\qquad\qquad\qquad\qquad\qquad\qquad\qquad\qquad\square$

Consider now the mean-field SDE starting from a random variable $\mathbf{W}_0^\star$ given by

$$\mathrm{d}\mathbf{W}_t^\star = (t+1)^{-\alpha} \left\{ h(\mathbf{W}_t^\star, \boldsymbol{\lambda}_t^\star)\mathrm{d}t + (\gamma^{1/(1-\alpha)}\Sigma(\mathbf{W}_t^\star, \boldsymbol{\lambda}_t^\star)/M)^{1/2}\mathrm{d}\mathbf{B}_t + \sqrt{2\eta}\mathrm{d}\tilde{\mathbf{B}}_t \right\} , \tag{S6}$$

where $\boldsymbol{\lambda}_t^\star$ is the distribution of $\mathbf{W}_t^\star$ and $(\mathbf{B}_t)_{t \geq 0}$ and $(\tilde{\mathbf{B}}_t)_{t \geq 0}$ are independent $p$ dimensional Brownian motions.

**Theorem S3.** *Let* $\beta = 1$. *Assume* **A**1. *Let* $(\mathbf{W}_0^k)_{k \in \mathbb{N}}$ *be a sequence of* $\mathbb{R}^p$-*valued random variables with distribution* $\mu_0 \in \mathscr{P}_2(\mathbb{R}^p)$ *and assume that for any* $N \in \mathbb{N}^\star$, $\mathbf{W}_0^{1:N} = \{\mathbf{W}_0^k\}_{k=1}^N$. *Then, for any* $m \in \mathbb{N}^\star$ *and* $T \geq 0$, *there exists* $C_{m,T} \geq 0$ *such that for any* $\alpha \in [0, 1)$, $M \in \mathbb{N}^\star$ *and* $N \in \mathbb{N}^\star$ *we have*

$$\mathbb{E}\left[\sup_{t \in [0,T]} \|\mathbf{W}_t^{1:m,N} - \mathbf{W}_t^{1:m,\star}\|^2\right] \leq C_{m,T}N^{-1} ,$$

*with* $(\mathbf{W}_t^{1:m,N}, \mathbf{W}_t^{1:m,\star}) = \{(\mathbf{W}_t^{k,N}, \mathbf{W}_t^{k,\star})\}_{k=1}^m$, $(\mathbf{W}_t^{1:N})$ *is the solution of* (S4) *starting from* $\mathbf{W}_0^{1:N}$, *and for any* $k \in \{1, \dots, N\}$, $\mathbf{W}_t^{k,\star}$ *is the solution of* (S6) *starting from* $\mathbf{W}_0^k$ *and Brownian motions* $(\mathbf{B}_t^k)_{t \geq 0}$ *and* $(\tilde{\mathbf{B}}_t^k)_{t \geq 0}$.

*Proof.* The proof is postponed to Section S4.4 $\qquad\qquad\qquad\qquad\qquad\qquad\qquad\qquad\square$

## S3 Technical results

In this section, we derive technical results needed to establish Theorem 1, Theorem 2, Theorem S2 and Theorem S3. In particular, we are interested in the regularity properties of the mean field $h$ and the diffusion matrix $\Sigma$ under **A**1. We recall that in this setting, for any $w \in \mathbb{R}^p$, $\mu \in \mathscr{P}(\mathbb{R}^p)$, $(x, y) \in \mathsf{X} \times \mathsf{Y}$, we have

$$h(w, \mu) = \tilde{h}(w, \mu) - \nabla V(w) ,$$

$$\text{with} \quad \tilde{h}(w, \mu) = -\int_{\mathsf{X} \times \mathsf{Y}} \partial_1 \ell\left(\int_{\mathbb{R}^p} F(\zeta, x)\,\mathrm{d}\mu(\zeta), y\right) \nabla_w F(w, x)\,\mathrm{d}\pi(x, y) ,$$

$$\xi(w,\mu,x,y) = -\tilde{h}(w,\mu) - \partial_1\ell\left(\int_{\mathbb{R}^p} F(\zeta,x)\,\mathrm{d}\mu(\zeta), y\right)\nabla_w F(w,x)\,,$$

$$\Sigma(w,\mu) = \int_{\mathsf{X}\times\mathsf{Y}}\{\xi\xi^\top\}(w,\mu,x,y)\mathrm{d}\pi(x,y)\,, \qquad \mathrm{S}(w,\mu) = \Sigma^{1/2}(w,\mu)\,. \tag{S7}$$

Note that by **A**1-(a), we obtain the following estimate used in the proof of the results of this Section: for any $\mathrm{y}, y \in \mathbb{R}$

$$|\partial_1\ell(\mathrm{y},y)| \le |\partial_1\ell(0,y)| + \Psi(y)\,|\mathrm{y}| \le 2\Psi(y)\max(1,|\mathrm{y}|)\,. \tag{S8}$$

In addition, note that under **A**1-(c), there exists $\mathrm{K} \ge 0$ such that for any $w \in \mathbb{R}^p$

$$\left\|\nabla^2 V(w)\right\| + \left\|\mathrm{D}^3 V(w)\right\| \le \mathrm{K}\,, \qquad \left\|\nabla V(w)\right\| \le \mathrm{K}(1 + \|w\|)\,. \tag{S9}$$

Let $G: \mathbb{R}^p \times \mathsf{X} \times \mathsf{Y} \to \mathbb{R}$ given for any $(x,y) \in \mathsf{X} \times \mathsf{Y}$ and $w \in \mathbb{R}^p$ by

$$G(w,x,y) = \{\Phi^4(x) + \Psi^2(y)\}F(w,x)\,. \tag{S10}$$

We now state our main regularity/boundedness proposition.

**Proposition S4.** *Assume* **A**1. *Then, there exists* $\mathrm{L} \ge 0$ *such that the following hold.*

*(a) For any $\mu_1, \mu_2 \in \mathscr{P}(\mathbb{R}^p)$ and $w_1, w_2 \in \mathbb{R}^p$ we have*

$$\|h(w_1,\mu_1) - h(w_2,\mu_2)\|$$
$$\le \mathrm{L}\left\{\|w_1 - w_2\| + \left(\int_{\mathsf{X}\times\mathsf{Y}}\|\mu_1[G(\cdot,x,y)] - \mu_2[G(\cdot,x,y)]\|^2\,\mathrm{d}\pi(x,y)\right)^{1/2}\right\}\,. \tag{S11}$$

*In addition, we have for any $\mu \in \mathscr{P}(\mathbb{R}^p)$ and $w \in \mathbb{R}^p$, $\|h(w,\mu)\| \le \mathrm{L}(1 + \|w\|)$ and $\|\bar{h}(w,\mu)\| \le \mathrm{L}$.*

*(b) For any $\mu_1, \mu_2 \in \mathscr{P}(\mathbb{R}^p)$, $w_1, w_2 \in \mathbb{R}^p$ and $i,j \in \{1,\dots,p\}$ we have*

$$|\mathrm{S}_{i,j}(w_1,\mu_1) - \mathrm{S}_{i,j}(w_2,\mu_2)|$$
$$\le \mathrm{L}\left\{\|w_1 - w_2\| + \left(\int_{\mathsf{X}\times\mathsf{Y}}\|\mu_1[G(\cdot,x,y)] - \mu_2[G(\cdot,x,y)]\|^2\,\mathrm{d}\pi(x,y)\right)^{1/2}\right\}\,. \tag{S12}$$

*In addition, we have for any $\mu \in \mathscr{P}(\mathbb{R}^p)$, $w \in \mathbb{R}^p$ and $i,j \in \{1,\dots,p\}$, $|\mathrm{S}_{i,j}(w,\mu)| \le \mathrm{L}$.*

*(c) For any $\mu \in \mathscr{P}(\mathbb{R}^p)$ and $w \in \mathbb{R}^p$, $\int_{\mathsf{X}\times\mathsf{Y}}\|\xi(w,\mu,x,y)\|^2\,\mathrm{d}\pi(x,y) \le p^2\mathrm{L}^2$.*

*Proof.* (a) First, we show that (S11) holds. Note that by the triangle inequality and (S7), we only need to consider $h \leftarrow \tilde{h}$ and $h \leftarrow V$. The case $h \leftarrow V$ is straightforward using (S9). We now deal with the first case. For any $w_1, w_2 \in \mathbb{R}^p$ and $\mu_1, \mu_2 \in \mathscr{P}(\mathbb{R}^p)$, consider the decomposition,

$$\|\tilde{h}(w_1,\mu_1) - \tilde{h}(w_2,\mu_2)\| \le \|\tilde{h}(w_1,\mu_1) - \tilde{h}(w_2,\mu_1)\| + \|\tilde{h}(w_2,\mu_1) - \tilde{h}(w_2,\mu_2)\|\,.$$

In what follows, we bound separately the two terms in the right-hand side. Using **A**1-(a), **A**1-(b), (S7) and (S8) we have for any $w_1, w_2 \in \mathbb{R}^p$ and $\mu_1 \in \mathscr{P}(\mathbb{R}^p)$

$$\|\tilde{h}(w_1,\mu_1) - \tilde{h}(w_2,\mu_1)\| \le \left\|\int_{\mathsf{X}\times\mathsf{Y}}\partial_1\ell(\mu_1[F(\cdot,x)],y)\nabla_w F(w_1,x)\,\mathrm{d}\pi(x,y)\right.$$
$$\left. - \int_{\mathsf{X}\times\mathsf{Y}}\partial_1\ell(\mu_1[F(\cdot,x)],y)\nabla_w F(w_2,x)\,\mathrm{d}\pi(x,y)\right\|$$
$$\le \int_{\mathsf{X}\times\mathsf{Y}}|\partial_1\ell(\mu_1[F(\cdot,x)],y)|\,\Phi(x)\mathrm{d}\pi(x,y)\,\|w_1 - w_2\|$$
$$\le \int_{\mathsf{X}\times\mathsf{Y}}\Psi(y)\Phi(x)\left(1 + |\mu_1[F(\cdot,x)]|\right)\mathrm{d}\pi(x,y)\,\|w_1 - w_2\|$$
$$\le 2\int_{\mathsf{X}\times\mathsf{Y}}\Psi(y)\Phi^2(x)\mathrm{d}\pi(x,y)\,\|w_1 - w_2\|\,. \tag{S13}$$

Using **A**1-(a), **A**1-(b), (S7) and the Cauchy-Schwarz inequality, we also have for any $w_1 \in \mathbb{R}^p$ and $\mu_1, \mu_2 \in \mathscr{P}(\mathbb{R}^p)$

$$
\begin{aligned}
&\|\tilde{h}(\mu_1, w_1) - \tilde{h}(\mu_2, w_1)\| \\
&\quad \leq \left\| \int_{\mathsf{X} \times \mathsf{Y}} \{\partial_1 \ell(\mu_1[F(\cdot, x)], y) \nabla_w F(w_1, x) - \partial_1 \ell(\mu_2[F(\cdot, x)], y) \nabla_w F(w_1, x)\} \, \mathrm{d}\pi(x, y) \right\| \\
&\quad \leq \int_{\mathsf{X} \times \mathsf{Y}} |\partial_1 \ell(\mu_1[F(\cdot, x)], y) - \partial_1 \ell(\mu_2[F(\cdot, x)], y)| \, \|\nabla_w F(w_1, x)\| \, \mathrm{d}\pi(x, y) \\
&\quad \leq \int_{\mathsf{X} \times \mathsf{Y}} \Psi(y) \|\mu_1[F(\cdot, x)] - \mu_2[F(\cdot, x)]\| \, \Phi(x) \mathrm{d}\pi(x, y) \\
&\quad \leq \left( \int_{\mathsf{X} \times \mathsf{Y}} \Psi^2(y) \Phi^2(x) \mathrm{d}\pi(x, y) \right)^{1/2} \left( \int_{\mathsf{X}} \|\mu_1[F(\cdot, x)] - \mu_2[F(\cdot, x)]\|^2 \, \mathrm{d}\pi(x) \right)^{1/2} . \quad \text{(S14)}
\end{aligned}
$$

Combining (S10), (S13), (S14), the fact that for any $a, b \geq 0$, $2ab \leq a^2 + b^2$ and **A**1-(d), we obtain that there exists $\mathsf{L}_1 \geq 0$ such that for any $\mu_1, \mu_2 \in \mathscr{P}(\mathbb{R}^p)$ and $w_1, w_2 \in \mathbb{R}^p$ we have

$$
\begin{aligned}
&\|\tilde{h}(w_1, \mu_1) - \tilde{h}(w_2, \mu_2)\| \\
&\quad \leq \mathsf{L}_1 \left\{ \|w_1 - w_2\| + \left( \int_{\mathsf{X} \times \mathsf{Y}} \|\mu_1[G(\cdot, x, y)] - \mu_2[G(\cdot, x, y)]\|^2 \, \mathrm{d}\pi(x, y) \right)^{1/2} \right\} .
\end{aligned}
$$

In addition, using **A**1-(b) and (S8), we have for any $w \in \mathbb{R}^p$, $\mu \in \mathscr{P}(\mathbb{R}^p)$, $x \in \mathsf{X}$ and $y \in \mathsf{Y}$

$$
|\partial_1 \ell(\mu[F(\cdot, x)], y)| \, \|\nabla_w F(w, x)\| \leq \Psi(y) \Phi(x)(1 + \Phi(x)) \leq 2\Psi(y)\Phi^2(x) . \quad \text{(S15)}
$$

Therefore, combining this result and (S7), we get that for any $w \in \mathbb{R}^p$ and $\mu \in \mathscr{P}(\mathbb{R}^p)$

$$
\|\tilde{h}(w, \mu)\| \leq \int_{\mathsf{X} \times \mathsf{Y}} 2\Psi(y)\Phi^2(x) \mathrm{d}\pi(x, y) .
$$

Using the fact that for any $a, b \geq 0$, $2ab \leq a^2 + b^2$ and **A**1-(d), there exists $\mathsf{L}_2 \geq 0$ such that for any $w \in \mathbb{R}^p$ and $\mu \in \mathscr{P}(\mathbb{R}^p)$,

$$
\|\tilde{h}(w, \mu)\| \leq \mathsf{L}_2 \quad \text{(S16)}
$$

(b) Second, we first show that there exists $\mathsf{L}_3 \geq 0$ such that for any $\mu \in \mathscr{P}(\mathbb{R}^p)$, $w \in \mathbb{R}^p$ and $i, j \in \{1, \ldots, p\}$, $|\mathsf{S}_{i,j}(w, \mu)| \leq \mathsf{L}$. Let $i, j \in \{1, \ldots, p\}$. We have for any $w \in \mathbb{R}^p$ and $\mu \in \mathscr{P}(\mathbb{R}^p)$

$$
|\mathsf{S}_{i,j}(w, \mu)| \leq \|\mathsf{S}(w, \mu)\| \leq \mathrm{Tr}^{1/2} \left( \Sigma(w, \mu) \right) . \quad \text{(S17)}
$$

Similarly to (S15), using (S7), (S16), the fact that for any $a, b \geq 0$, $(a + b)^2 \leq 2(a^2 + b^2)$ and the Cauchy-Schwarz inequality, we get for any $w \in \mathbb{R}^p$ and $\mu \in \mathscr{P}(\mathbb{R}^p)$

$$
\mathrm{Tr}\left( \Sigma(w, \mu) \right) \leq \int_{\mathsf{X} \times \mathsf{Y}} \|\xi(w, \mu, x, y)\|^2 \, \mathrm{d}\pi(x, y) \leq 2 \int_{\mathsf{X} \times \mathsf{Y}} \{\mathsf{L}_2^2 + 2\Psi^2(y)\Phi^4(x)\} \mathrm{d}\pi(x, y) . \quad \text{(S18)}
$$

Combining (S17), (S18) and **A**1-(d), there exists $\mathsf{L}_3 \geq 0$ such that for any $w \in \mathbb{R}^p$ and $\mu \in \mathscr{P}(\mathbb{R}^p)$, $\max_{1 \leq i, j \leq p} |\mathsf{S}_{i,j}(w, \mu)| \leq \mathsf{L}_3$.

We now show that (S12) holds. For any $w_1, w_2 \in \mathbb{R}^p$, $\mu_1, \mu_2 \in \mathscr{P}(\mathbb{R}^p)$ define $\varphi_\Sigma : [0, 1] \to \mathbb{S}_p(\mathbb{R})$ for any $t \in [0, 1]$ by

$$
\varphi_\Sigma(t) = \Sigma(tw_1 + (1 - t)w_2, t\mu_1 + (1 - t)\mu_2) . \quad \text{(S19)}
$$

For ease of notation, the dependency of $\varphi_\Sigma$ with respect to $w_1, w_2 \in \mathbb{R}^p$ and $\mu_1, \mu_2 \in \mathscr{P}(\mathbb{R}^p)$ is omitted. In what follows, we show that for any $w_1, w_2 \in \mathbb{R}^p$, $\mu_1, \mu_2 \in \mathscr{P}(\mathbb{R}^p)$, $\varphi_\Sigma \in \mathrm{C}^2([0, 1], \mathbb{S}_p(\mathbb{R}))$ and that there exists $\mathsf{L}_4 \geq 0$ such that for any $t \in [0, 1]$

$$
\|\varphi_\Sigma''(t)\| \leq \mathsf{L}_4 \left\{ \|w_1 - w_2\| + \left( \int_{\mathsf{X} \times \mathsf{Y}} \|\mu_1[G(\cdot, x, y)] - \mu_2[G(\cdot, x, y)]\|^2 \, \mathrm{d}\pi(x, y) \right)^{1/2} \right\}^2 ,
$$

which will conclude the proof of (S12) upon using a straightforward adaptation of [5, Lemma 3.2.3, Theorem 5.2.3]. We conclude the proof of Proposition S4 upon letting $\mathsf{L} = \max(\mathsf{L}_1, \mathsf{L}_2, \mathsf{L}_3, \mathsf{L}_4)$.

For any $t \in [0,1]$, let $\mu_t = \mu_1 + t(\mu_2 - \mu_1) \in \mathscr{P}(\mathbb{R}^p)$ and $w_t = w_1 + t(w_2 - w_1) \in \mathbb{R}^p$ and for any $(x,y) \in \mathsf{X} \times \mathsf{Y}$ define

$$
\mathrm{f}(t,x,y) = \partial_1 \ell(\mu_t[F(\cdot,x)],y) \nabla_w F(w_t,x) ,
$$
$$
\tilde{\mathrm{f}}(t,x,y) = \xi(w_t,\mu_t,x,y) = \int_{\mathsf{X} \times \mathsf{Y}} \mathrm{f}(t,x,y) \mathrm{d}\pi(x,y) - \mathrm{f}(t,x,y) . \tag{S20}
$$

The rest of the proof consists in showing that $\varphi_\Sigma$ is twice differentiable with dominated derivatives using the Lebesgue convergence theorem.

By (S7), (S15) and (S16), we get that for any $w_1, w_2 \in \mathbb{R}^p$, $\mu_1, \mu_2 \in \mathscr{P}(\mathbb{R}^p)$, $(x,y) \in \mathsf{X} \times \mathsf{Y}$ and $t \in [0,1]$

$$
\|\mathrm{f}(t,x,y)\| \le 2\Psi(y)\Phi^2(x) , \qquad \|\tilde{\mathrm{f}}(t,x,y)\| \le \mathrm{L}_2 + 2\Psi(y)\Phi^2(x) . \tag{S21}
$$

Using (S20), **A**1-(a) and **A**1-(b), we have that for any $(x,y) \in \mathsf{X} \times \mathsf{Y}$, $\mathrm{f}(\cdot,x,y) \in \mathrm{C}^1([0,1],\mathbb{R}^p)$ and for any $w_1, w_2 \in \mathbb{R}^p$, $\mu_1, \mu_2 \in \mathscr{P}(\mathbb{R}^p)$, $(x,y) \in \mathsf{X} \times \mathsf{Y}$ and $t \in [0,1]$

$$
\partial_1 \mathrm{f}(t,x,y) = \partial_1^2 \ell(\mu_t[F(\cdot,x)],y) \nabla_w F(w_t,x) (\mu_2[F(\cdot,x)] - \mu_1[F(\cdot,x)])
$$
$$
+ \partial_1 \ell(\mu_t[F(\cdot,x)],y) \nabla_w^2 F(w_t,x)(w_2 - w_1) . \tag{S22}
$$

Using **A**1-(a), **A**1-(b), (S10) and (S8), we get that for any $(x,y) \in \mathsf{X} \times \mathsf{Y}$ and $t \in [0,1]$

$$
\|\partial_1 \mathrm{f}(t,x,y)\| \le 3\Psi(y)\Phi^2(x) \left( \|w_2 - w_1\| + \|\mu_1[F(\cdot,x)] - \mu_2[F(\cdot,x)]\| \right) , \tag{S23}
$$

Similarly, using (S22), **A**1-(a) and **A**1-(b), we have that for any $(x,y) \in \mathsf{X} \times \mathsf{Y}$, $\mathrm{f}(\cdot,x,y) \in \mathrm{C}^2([0,1],\mathbb{R}^p)$ and for any $w_1, w_2 \in \mathbb{R}^p$, $\mu_1, \mu_2 \in \mathscr{P}(\mathbb{R}^p)$, $(x,y) \in \mathsf{X} \times \mathsf{Y}$ and $t \in [0,1]$

$$
\partial_1^2 \mathrm{f}(t,x,y) = \partial_1^3 \ell(\mu_t[F(\cdot,x)],y) \nabla_w F(w_t,x) (\mu_2[F(\cdot,x)] - \mu_1[F(\cdot,x)])^2
$$
$$
+ 2\partial_1^2 \ell(\mu_t[F(\cdot,x)],y) \nabla_w^2 F(w_t,x)(w_2 - w_1) (\mu_2[F(\cdot,x)] - \mu_1[F(\cdot,x)])
$$
$$
+ \partial_1 \ell(\mu_t[F(\cdot,x)],y) \mathrm{D}_w^3 F(w_t,x)(w_2 - w_1)^{\otimes 2} .
$$

Using **A**1-(a), **A**1-(b) and (S8) and that for any $a,b \ge 0$, $2ab \le a^2 + b^2$, we get that for any $(x,y) \in \mathsf{X} \times \mathsf{Y}$ and $t \in [0,1]$

$$
\left\|\partial_1^2 \mathrm{f}(t,x,y)\right\| \le 5\Psi(y)\Phi^2(x) \left( \|w_2 - w_1\|^2 + \|\mu_1[F(\cdot,x)] - \mu_2[F(\cdot,x)]\|^2 \right) . \tag{S24}
$$

Combining (S20), (S23), (S24), **A**1-(d) and the dominated convergence theorem, we get that for any $(x,y) \in \mathsf{X} \times \mathsf{Y}$, $\tilde{\mathrm{f}}(\cdot,x,y) \in \mathrm{C}^2([0,1],\mathbb{R}^p)$. In addition, using (S20), (S21), (S23), (S24), the Cauchy-Schwarz inequality and the fact that for any $a,b \ge 0$, $2ab \le a^2 + b^2$, there exists $C \ge 0$, such that for any $w_1, w_2 \in \mathbb{R}^p$, $\mu_1, \mu_2 \in \mathscr{P}(\mathbb{R}^p)$, $(x,y) \in \mathsf{X} \times \mathsf{Y}$ and $t \in [0,1]$

$$
\|\tilde{\mathrm{f}}(t,x,y)\| \le C \left( \Phi^4(x) + \Psi^2(y) \right) ,
$$
$$
\|\partial_1 \tilde{\mathrm{f}}(t,x,y)\| \le C \left( \Phi^4(x) + \Psi^2(y) \right) \chi(w_1,w_2,\mu_1,\mu_2,x) ,
$$
$$
\|\partial_1^2 \tilde{\mathrm{f}}(t,x,y)\| \le C \left( \Phi^4(x) + \Psi^2(y) \right) \chi^2(w_1,w_2,\mu_1,\mu_2,x) , \tag{S25}
$$

where

$$
\chi(w_1,w_2,\mu_1,\mu_2,x) = \|w_1 - w_2\|
$$
$$
+ \|\mu_1[F(\cdot,x)] - \mu_2[F(\cdot,x)]\| + \left( \int_{\mathsf{X} \times \mathsf{Y}} \|\mu_1[G(\cdot,\tilde{x},\tilde{y})] - \mu_2[G(\cdot,\tilde{x},\tilde{y})]\|^2 \, \mathrm{d}\pi(\tilde{x},\tilde{y}) \right)^{1/2} .
$$

Using (S19) and (S7), we have that for any $w_1, w_2 \in \mathbb{R}^p$, $\mu_1, \mu_2 \in \mathscr{P}(\mathbb{R}^p)$, $t \in [0,1]$

$$
\varphi_\Sigma(t) = \int_{\mathsf{X} \times \mathsf{Y}} \tilde{\mathrm{f}}(t,x,y)\tilde{\mathrm{f}}(t,x,y)^\top \mathrm{d}\pi(x,y) .
$$

Combining this result, (S25) and **A**1-(d) we get that for any $w_1, w_2 \in \mathbb{R}^p$ and $\mu_1, \mu_2 \in \mathscr{P}(\mathbb{R}^p)$, $\varphi_\Sigma \in \mathrm{C}^2([0,1], \mathbb{S}_p(\mathbb{R}))$ and, using the Cauchy-Schwarz inequality, there exist $C_1, C_2 \ge 0$ such that for any $w_1, w_2 \in \mathbb{R}^p$ and $\mu_1, \mu_2 \in \mathscr{P}(\mathbb{R}^p)$, $t \in [0,1]$ and $\mathrm{u} \in \mathbb{R}^p$ with $\|\mathrm{u}\| = 1$, we have

$$
\langle \mathrm{u}, \varphi_\Sigma''(t)\mathrm{u} \rangle = \int_{\mathsf{X} \times \mathsf{Y}} \partial_1^2 \left( \langle \mathrm{u}, \tilde{\mathrm{f}}(t,x,y) \rangle^2 \right) \mathrm{d}\pi(x,y)
$$

$$\leq 2 \int_{\mathsf{X} \times \mathsf{Y}} \|\partial_1 \tilde{\mathrm{f}}(t,x,y)\|^2 \mathrm{d}\pi(x,y) + 2 \int_{\mathsf{X} \times \mathsf{Y}} \|\partial_1^2 \tilde{\mathrm{f}}(t,x,y)\| \|\tilde{\mathrm{f}}(t,x,y)\| \mathrm{d}\pi(x,y)$$

$$\leq C_1 \int_{\mathsf{X} \times \mathsf{Y}} \left( \Phi^8(x) + \Psi^4(y) \right) \chi^2(w_1, w_2, x, y) \mathrm{d}\pi(x,y)$$

$$\leq C_2 \left\{ \|w_1 - w_2\| + \left( \int_{\mathsf{X}} \|\mu_1[G(\cdot, x, y)] - \mu_2[G(\cdot, x, y)]\|^2 \, \mathrm{d}\pi(x,y) \right)^{1/2} \right\}^2 ,$$

Therefore, we get that for any $w_1, w_2 \in \mathbb{R}^p$, $\mu_1, \mu_2 \in \mathscr{P}(\mathbb{R}^p)$, $t \in [0,1]$

$$\|\varphi_\Sigma''(t)\| = \sup_{\mathrm{u} \in \mathbb{R}^p, \|\mathrm{u}\|=1} \langle \mathrm{u}, \varphi_\Sigma''(t)\mathrm{u} \rangle$$

$$\leq C \left\{ \|w_1 - w_2\| + \left( \int_{\mathsf{X}} \|\mu_1[G(\cdot, x, y)] - \mu_2[G(\cdot, x, y)]\|^2 \, \mathrm{d}\pi(x,y) \right)^{1/2} \right\}^2 .$$

Combining this result and a straightforward adaptation of [5, Lemma 3.2.3, Theorem 5.2.3] we obtain that for any $w_1, w_2 \in \mathbb{R}^p$, $\mu_1, \mu_2 \in \mathscr{P}(\mathbb{R}^p)$

$$|\mathrm{S}_{i,j}(w_1, \mu_1) - \mathrm{S}_{i,j}(w_2, \mu_2)| \leq \mathrm{L}_4 \left\{ \|w_1 - w_2\| + \left( \int_{\mathsf{X}} \|\mu_1[G(\cdot, x, y)] - \mu_2[G(\cdot, x, y)]\|^2 \, \mathrm{d}\pi(x,y) \right)^{1/2} \right\} ,$$

with $\mathrm{L}_4 = \sqrt{2C} p$.

(c) Using (S7), we have for any $w \in \mathbb{R}^p$ and $\mu \in \mathscr{P}(\mathbb{R}^p)$

$$\int_{\mathsf{X} \times \mathsf{Y}} \|\xi(w, \mu, x, y)\|^2 \, \mathrm{d}\pi(x,y) = \int_{\mathsf{X} \times \mathsf{Y}} \mathrm{Tr} \left( \xi \xi^\top (w, \mu, x, y) \right) \mathrm{d}\pi(x,y) = \sum_{i,j=1}^p |\mathrm{S}_{i,j}(w,\mu)|^2 \leq p^2 \mathrm{L}^2 .$$

$\square$

## S4 Quantitative propagation of chaos

### S4.1 Existence of strong solutions to the particle SDE

In this section, for two functions $A, B : \bigcup_{N \in \mathbb{N}^\star} \left\{ \{1, \ldots, N\} \times \mathbb{R}_+ \times (\mathbb{R}^p)^2 \times (\mathscr{P}_2(\mathbb{R}^p))^2 \right\} \to \mathbb{R}$, the notation $A_N(k, t, w_1, w_2, \mu_1, \mu_2) \lesssim B_N(k, t, w_1, w_2, \mu_1, \mu_2)$ stands for the statement that there exists $C \geq 0$ such that for any $N \in \mathbb{N}^\star$, $k \in \{1, \ldots, N\}$, $t \in \mathbb{R}_+$, $w_1, w_2 \in \mathbb{R}^p$, $\mu_1, \mu_2 \in \mathscr{P}_2(\mathbb{R}^p)$, $A_N(k, t, w_1, w_2, \mu_1, \mu_2) \leq C B_N(k, t, w_1, w_2, \mu_1, \mu_2)$, where $A_N$ and $B_N$ are the restrictions of $A$ and $B$ to $\{1, \ldots, N\} \times \mathbb{R}_+ \times (\mathbb{R}^p)^2 \times (\mathscr{P}_2(\mathbb{R}^p))^2$.

We consider for $N \in \mathbb{N}^\star$, $p$ dimensional particle system $(\mathbf{W}_t^{1:N})_{t \geq 0}$ associated with the SDE: for any $k \in \{1, \ldots, N\}$

$$\mathrm{d}\mathbf{W}_t^{k,N} = b_N(t, \mathbf{W}_t^{k,N}, \boldsymbol{\nu}_t^N)\mathrm{d}t + \sigma_N(t, \mathbf{W}_t^{k,N}, \boldsymbol{\nu}_t^N)\mathrm{d}\mathbf{B}_t^k , \qquad \boldsymbol{\nu}_t^N = (1/N) \sum_{k=1}^N \delta_{\mathbf{W}_t^{k,N}} , \quad (\text{S26})$$

where $(\mathbf{B}_t^k)_{k \in \mathbb{N}^\star}$ are independent $r$-dimensional Brownian motions and where $(b_N)_{N \in \mathbb{N}^\star}$ and $(\sigma_N)_{N \in \mathbb{N}^\star}$ are family of measurable functions such that for any $N \in \mathbb{N}^\star$, $b_N : \mathbb{R}_+ \times \mathbb{R}^p \times \mathscr{P}_2(\mathbb{R}^p) \to \mathbb{R}^p$ and $\sigma_N : \mathbb{R}_+ \times \mathbb{R}^p \times \mathscr{P}_2(\mathbb{R}^p) \to \mathbb{R}^{p \times r}$. We make the following assumption ensuring the existence and uniqueness of solutions of (S26) for any $N \in \mathbb{N}^\star$. Consider in the sequel a measurable space $(\mathsf{Z}, \mathcal{Z})$ and a probability measure $\pi_\mathsf{Z}$ on this space.

**B1.** *There exist a measurable function* $\mathrm{g} : \mathbb{R}^p \times \mathsf{Z} \to \mathbb{R}$, $\mathrm{M}_1 \geq 0$ *and* $\mu_0 \in \mathscr{P}_2(\mathbb{R}^p)$ *such that for any* $N \in \mathbb{N}^\star$, *the following hold.*

*(a) For any* $w_1, w_2 \in \mathbb{R}^p$ *and* $z \in \mathsf{Z}$ *we have*

$$\|\mathrm{g}(w_1, z) - \mathrm{g}(w_2, z)\| \leq \zeta(z) \|w_1 - w_2\| , \quad \|\mathrm{g}(w_1, z)\| \leq \zeta(z) , \quad with \int_\mathsf{Z} \zeta^2(z) \mathrm{d}\pi_\mathsf{Z}(z) < +\infty .$$

(b) $b_N \in C(\mathbb{R}_+ \times \mathbb{R}^p \times \mathscr{P}_2(\mathbb{R}^p), \mathbb{R}^p)$ and $\sigma_N \in C(\mathbb{R}_+ \times \mathbb{R}^p \times \mathscr{P}_2(\mathbb{R}^p), \mathbb{R}^{p \times r})$.

(c) For any $w_1, w_2 \in \mathbb{R}^p$ and $\mu_1, \mu_2 \in \mathscr{P}_2(\mathbb{R}^p)$

$$\sup_{t \geq 0} \{\|b_N(t, w_1, \mu_1) - b_N(t, w_2, \mu_2)\| + \|\sigma_N(t, w_1, \mu_1) - \sigma_N(t, w_2, \mu_2)\|\}$$

$$\leq \mathtt{M}_1 \left\{ \|w_1 - w_2\| + \left( \int_\mathsf{Z} |\mu_1[\mathrm{g}(\cdot, z)] - \mu_2[\mathrm{g}(\cdot, z)]|^2 \, \mathrm{d}\pi_\mathsf{Z}(z) \right)^{1/2} \right\},$$

$$\sup_{t \geq 0} \{\|b_N(t, 0, \mu_0)\| + \|\sigma_N(t, 0, \mu_0)\|\} \leq \mathtt{M}_1 .$$

**B2.** *There exist* $\mathtt{M}_2 \geq 0$, $\kappa > 0$, $b \in C(\mathbb{R}_+ \times \mathbb{R}^p \times \mathscr{P}_2(\mathbb{R}^p), \mathbb{R}^p)$ *and* $\sigma \in C(\mathbb{R}_+ \times \mathbb{R}^p \times \mathscr{P}_2(\mathbb{R}^p), \mathbb{R}^{p \times r})$ *such that*

$$\sup_{t \geq 0, w \in \mathbb{R}^p, \mu \in \mathscr{P}_2(\mathbb{R}^p)} \{\|b_N(t, w, \mu) - b(t, w, \mu)\| + \|\sigma_N(t, w, \mu) - \sigma(t, w, \mu)\|\} \leq \mathtt{M}_2 N^{-\kappa} .$$

Note that under **B**1, we have the following estimate which will be used in our next result,

$$\|b_N(t, w, \mu)\| + \|\sigma_N(t, w, \mu)\| \lesssim \left[ 1 + \|w\| + \left( \int_{\mathbb{R}^p} (1 + \|\tilde{w}\|^2) \mathrm{d}\mu(\tilde{w}) \right)^{1/2} \right], \qquad \text{(S27)}$$

$$\sup_{t \geq 0} \{\|b_N(t, w_1, \mu_1) - b_N(t, w_2, \mu_2)\| + \|\sigma_N(t, w_1, \mu_1) - \sigma_N(t, w_2, \mu_2)\|\}$$

$$\lesssim \|w_1 - w_2\| + \mathscr{W}_2(\mu_1, \mu_2) .$$

**Theorem S5.** *Assume* **B**1*. Then for any* $N \in \mathbb{N}^\star$, (S26) *admits a unique strong solution. If in addition, there exists* $m \geq 1$ *such that* $\sup_{N \in \mathbb{N}^\star} \sup_{k \in \{1, \ldots, N\}} \mathbb{E}[\|\mathbf{W}_0^{k,N}\|^{2m}] < +\infty$, *then for any* $T \geq 0$, *there exists* $C \geq 0$ *such that*

$$\sup_{N \in \mathbb{N}^\star} \sup_{k \in \{1, \ldots, N\}} \mathbb{E} \left[ \sup_{t \in [0, T]} \left\| \mathbf{W}_t^{k,N} \right\|^{2m} \right] \leq C .$$

*Proof.* First, we show that for any $N \in \mathbb{N}^\star$, (S26) admits a unique strong solution. Let $\tilde{b}_N : \mathbb{R}_+ \times (\mathbb{R}^p)^N \to (\mathbb{R}^p)^N$ and $\tilde{\sigma}_N : \mathbb{R}_+ \times (\mathbb{R}^p)^N \to (\mathbb{R}^{p \times r})^N$ given, setting $\nu^{N,w} = (1/N) \sum_{j=1}^N \delta_{w^{j,N}}$ for any $t \geq 0$ and $w^{1:N} \in (\mathbb{R}^p)^N$, by

$$\tilde{b}_N(t, w^{1:N}) = \left( b_N \left( t, w^{k,N}, \nu^{N,w} \right) \right)_{k \in \{1, \ldots, N\}}, \quad \tilde{\sigma}_N(t, w^{1:N}) = \left( \sigma_N \left( t, w^{k,N}, \nu^{N,w} \right) \right)_{k \in \{1, \ldots, N\}} .$$

Let $w_1^{1:N}, w_2^{1:N} \in (\mathbb{R}^p)^N$. Using **B**1, Proposition S1 and that for any $a, b \geq 0$, $(a + b)^{1/2} \leq a^{1/2} + b^{1/2}$, we have

$$\|b_N(t, w_1^{k,N}, \nu^{N,w_1}) - b_N(t, w_2^{k,N}, \nu^{N,w_2})\| \lesssim \|w_1^{k,N} - w_2^{k,N}\| + \mathscr{W}_2(\nu^{N,w_1}, \nu^{N,w_2})$$

$$\lesssim \|w_1^{k,N} - w_2^{k,N}\| + (N^{-1} \sum_{j=1}^N \|w_1^{j,N} - w_2^{j,N}\|^2)^{1/2} \lesssim \|w_1^{1:N} - w_2^{1:N}\| .$$

Similarly, we have $\|\sigma_N(t, w_1^{k,N}, \nu^{N,w_1}) - \sigma_N(t, w_2^{k,N}, \nu^{N,w_2})\| \lesssim \|w_1^{1:N} - w_2^{1:N}\|$. Therefore, we obtain that for any $N \in \mathbb{N}^\star$, $\tilde{b}_N$ and $\tilde{\sigma}_N$ are Lipschitz-continuous and using [6, Theorem 2.9], we get that there exists a unique strong solution to (S26). Let $m \geq 1$ and assume that $\sup_{N \in \mathbb{N}^\star} \sup_{k \in \{1, \ldots, N\}} \mathbb{E}[\|\mathbf{W}_0^{k,N}\|^{2m}] < +\infty$, we now show that for any $T \geq 0$, there exists $C \geq 0$ such that

$$\sup_{t \in [0, T]} \sup_{N \in \mathbb{N}^\star} \sup_{k \in \{1, \ldots, N\}} \mathbb{E} \left[ \left\| \mathbf{W}_t^{k,N} \right\|^{2m} \right] \leq C .$$

Let $V_m : \mathbb{R}^p \to \mathbb{R}_+$ given for any $w \in \mathbb{R}^p$ by $V_m(w) = 1 + \|w\|^{2m}$. For any $w \in \mathbb{R}^p$ we have

$$\|\nabla V_m(w)\| = 2m \|w\|^{2m-1}, \qquad \left\| \nabla^2 V_m(w) \right\| \leq 2m(2m - 1) \|w\|^{2m-2} .$$

Combining this result with (S27), the Cauchy-Schwarz inequality and the fact that for any $a, b \geq 0$ and $n_1, n_2 \in \mathbb{N}$, $a^{n_1} b^{n_2} \leq a^{n_1+n_2} + b^{n_1+n_2}$, we get that

$$|\langle \nabla V_m(w), b_N(t, w, \mu) \rangle| + \left| \langle \nabla^2 V_m(w), \sigma_N \sigma_N^\top(t, w, \mu) \rangle \right|$$

$$\lesssim \left[1 + \|w\| + \left(\int_{\mathbb{R}^p}(1+\|\tilde{w}\|^2)\mathrm{d}\mu(\tilde{w})\right)^{1/2}\right]\|\nabla V_m(w)\|$$

$$+ \left[1 + \|w\| + \left(\int_{\mathbb{R}^p}(1+\|\tilde{w}\|^2)\mathrm{d}\mu(\tilde{w})\right)^{1/2}\right]^2 \|\nabla^2 V_m(w)\|$$

$$\lesssim \left[1 + \|w\| + \left(\int_{\mathbb{R}^p}(1+\|\tilde{w}\|^2)\mathrm{d}\mu(\tilde{w})\right)^{1/2}\right]\|w\|^{2m-1}$$

$$+ \left[1 + \|w\|^2 + \int_{\mathbb{R}^p}(1+\|\tilde{w}\|^2)\mathrm{d}\mu(\tilde{w})\right]\|w\|^{2m-2}$$

$$\lesssim 1 + \|w\|^{2m} + \left(\int_{\mathbb{R}^p}(1+\|\tilde{w}\|^2)\mathrm{d}\mu(\tilde{w})\right)^m \lesssim 1 + \|w\|^{2m} + \int_{\mathbb{R}^p}(1+\|\tilde{w}\|^{2m})\mathrm{d}\mu(\tilde{w}) \,.$$
$$(S28)$$

Now let $\tau_n^N = \inf\{t \geq 0 \ : \ \|\mathbf{W}_t^{k,N}\| \geq n \text{ for some } k \in \{1,\dots,N\}\}$. Using Itô's lemma, (S28) and (S26), we have

$$\mathbb{E}\left[V_m(\mathbf{W}_{t\wedge\tau_n^N}^{k,N})\right] = \mathbb{E}\left[V_m(\mathbf{W}_{0\wedge\tau_n^N}^{k,N})\right] + \mathbb{E}\left[\int_0^{t\wedge\tau_n^N}\left\langle\nabla V_m(\mathbf{W}_s^{k,N}), b_N\left(s,\mathbf{W}_s^{k,N},\boldsymbol{\nu}_s^N\right)\right\rangle\mathrm{d}s\right]$$

$$+ (1/2)\mathbb{E}\left[\int_0^{t\wedge\tau_n^N}\left\langle\nabla^2 V_m(\mathbf{W}_s^{k,N}),\sigma_N\sigma_N^\top\left(s,\mathbf{W}_s^{k,N},\boldsymbol{\nu}_s^N\right)\right\rangle\mathrm{d}s\right]$$

$$\lesssim \mathbb{E}\left[V_m(\mathbf{W}_{0\wedge\tau_n^N}^{k,N})\right] + \mathbb{E}\left[\int_0^{t\wedge\tau_n^N}\left\{V_m(\mathbf{W}_s^{k,N}) + (1/N)\sum_{j=1}^N V_m(\mathbf{W}_s^{j,N})\right\}\mathrm{d}s\right]$$

Using Fatou's lemma, since almost surely $\tau_n^N \to +\infty$ as $n \to +\infty$, we get that

$$\mathbb{E}\left[V_m(\mathbf{W}_t^{k,N}) + (1/N)\sum_{j=1}^N V_m(\mathbf{W}_t^{j,N})\right]$$

$$\lesssim \mathbb{E}\left[V_m(\mathbf{W}_0^{k,N}) + (1/N)\sum_{j=1}^N V_m(\mathbf{W}_0^{j,N})\right] + \int_0^t \mathbb{E}\left[V_m(\mathbf{W}_s^{k,N}) + (1/N)\sum_{j=1}^N V_m(\mathbf{W}_s^{j,N})\right]\mathrm{d}s \,.$$

Using Grönwall's lemma, we get that for any $T \geq 0$, there exists $C \geq 0$ such that

$$\sup_{t\in[0,T]}\sup_{N\in\mathbb{N}^\star}\sup_{k\in\{1,\dots,N\}}\mathbb{E}\left[\left\|\mathbf{W}_t^{k,N}\right\|^{2m}\right] \leq C \,.$$

We now show that there exists $C \geq 0$ such that

$$\sup_{N\in\mathbb{N}^\star}\sup_{k\in\{1,\dots,N\}}\mathbb{E}\left[\sup_{t\in[0,T]}\left\|\mathbf{W}_t^{k,N}\right\|^{2m}\right] \leq C \,.$$

Using Jensen's inequality, Burkholder-Davis-Gundy's inequality [7, IV.42], (S27) and the fact that for any $(a_j)_{j\in\{1,\dots,M\}}$ and $r \geq 1$ such that $a_j \geq 0$, $(\sum_{j=1}^M a_j)^r \leq M^{r-1}\sum_{j=1}^M a_j^r$ we get for any $m \in \mathbb{N}^\star$

$$\mathbb{E}\left[\sup_{t\in[0,T]}\left\|\mathbf{W}_t^{k,N}\right\|^{2m}\right]$$

$$\lesssim \mathbb{E}\left[\sup_{t\in[0,T]}\left\|\int_0^t b_N(s,\mathbf{W}_s^{k,N},\boldsymbol{\nu}_s^N)\mathrm{d}s\right\|^{2m}\right] + \mathbb{E}\left[\sup_{t\in[0,T]}\left\|\int_0^t \sigma_N^{1/2}(s,\mathbf{W}_s^{k,N},\boldsymbol{\nu}_s^N)\mathrm{d}\mathbf{B}_s\right\|^{2m}\right]$$

$$\lesssim \mathbb{E}\left[\int_0^T \left\|b_N(s, \mathbf{W}_s^{k,N}, \boldsymbol{\nu}_s^N)\right\|^{2m} \mathrm{d}s\right] + \mathbb{E}\left[\left(\int_0^T \mathrm{Tr}(\sigma_N \sigma_N^\top(s, \mathbf{W}_s^{k,N}, \boldsymbol{\nu}_s^N))\mathrm{d}s\right)^m\right]$$

$$\lesssim \int_0^T \left\{\mathbb{E}\left[\left\|b_N(s, \mathbf{W}_s^{k,N}, \boldsymbol{\nu}_s^N)\right\|^{2m}\right] + \mathbb{E}\left[\left\|\sigma_N(s, \mathbf{W}_s^{k,N}, \boldsymbol{\nu}_s^N)\right\|^{2m}\right]\right\}\mathrm{d}s$$

$$\lesssim \int_0^T \left\{1 + \mathbb{E}\left[\left\|\mathbf{W}_s^{k,N}\right\|^{2m}\right] + \mathbb{E}\left[\int_{\mathbb{R}^p}(1 + \|\tilde{w}\|^{2m})\mathrm{d}\boldsymbol{\nu}_s^N(\tilde{w})\right]\right\}\mathrm{d}s$$

$$\lesssim \int_0^T \left\{1 + \mathbb{E}\left[\left\|\mathbf{W}_s^{k,N}\right\|^{2m}\right] + (1/N)\sum_{j=1}^N \mathbb{E}\left[\left\|\mathbf{W}_s^{j,N}\right\|^{2m}\right]\right\}\mathrm{d}s$$

$$\lesssim 1 + \sup_{N \in \mathbb{N}^\star} \sup_{j \in \{1,\ldots,N\}} \sup_{t \in [0,T]} \mathbb{E}\left[\left\|\mathbf{W}_s^{j,N}\right\|^{2m}\right] ,$$

which concludes the proof. $\qquad\square$

### S4.2 Existence of solutions to the mean-field SDE

The following result is based on [8, Theorem 1.1] showing, under **B**1 and **B**2, the existence of strong solutions and pathwise uniqueness for non-homogeneous McKean-Vlasov SDE with non-constant covariance matrix:

$$\mathrm{d}\mathbf{W}_t^\star = b(t, \mathbf{W}_t^\star, \boldsymbol{\lambda}_t^\star)\mathrm{d}t + \sigma(t, \mathbf{W}_t^\star, \boldsymbol{\lambda}_t^\star)\mathrm{d}\mathbf{B}_t , \qquad (S29)$$

where $b$ and $\sigma$ are given in **B**2 and where for any $t \geq 0$, $\mathbf{W}_t^\star$ has distribution $\boldsymbol{\lambda}_t^\star \in \mathscr{P}_2(\mathbb{R}^p)$, $(\mathbf{B}_t)_{t \geq 0}$ is a $r$ dimensional Brownian motion and $\mathbf{W}_0^\star$ has distribution $\mu_0 \in \mathscr{P}_2(\mathbb{R}^p)$.

**Proposition S6.** *Assume* **B**1 *and* **B**2. *Let* $\mu_0 \in \mathscr{P}_2(\mathbb{R}^p)$. *Then, there exists a* $(\mathcal{F}_t)_{t \geq 0}$-*adapted continuous process* $(\mathbf{W}_t^\star)_{t \geq 0}$ *which is the unique strong solution of* (S29) *satisfying for any* $T \geq 0$, $\sup_{t \in [0,T]} \mathbb{E}[\|\mathbf{W}_t^\star\|^2] < +\infty$.

*Proof.* Let $\delta \geq 0$ and $\mu_0 \in \mathscr{P}_2(\mathbb{R}^p)$. Note that we only need to show that (S29) admits a strong solution up to $\bar{\delta} > 0$. First, using [6, Theorem 2.9], note that for any $(\boldsymbol{\mu}_t)_{t \in [0,\delta]} \in \mathscr{C}_{2,\delta}^p$ the SDE,

$$\mathrm{d}\mathbf{W}_t^{\boldsymbol{\mu}} = b(t, \mathbf{W}_t^{\boldsymbol{\mu}}, \boldsymbol{\mu}_t)\mathrm{d}t + \sigma(t, \mathbf{W}_t^{\boldsymbol{\mu}}, \boldsymbol{\mu}_t)\mathrm{d}\mathbf{B}_t ,$$

admits a unique strong solution, since for any $t \in [0, \delta]$ and $w_1, w_2 \in \mathbb{R}^p$

$$\|b(t, w_1, \boldsymbol{\mu}_t) - b(t, w_2, \boldsymbol{\mu}_t)\| + \|\sigma(t, w_1, \boldsymbol{\mu}_t) - \sigma(t, w_2, \boldsymbol{\mu}_t)\| \leq \mathtt{M}_1 \|w_1 - w_2\| . \qquad (S30)$$

In addition, $\sup_{t \in [0,\delta]} \mathbb{E}[\|\mathbf{W}_t^{\boldsymbol{\mu}}\|^2] < +\infty$.

In the rest of the proof, the strategy is to adapt the well-known Cauchy-Lipschitz approach using the Picard fixed point theorem. More precisely, we define below for $\delta > 0$ small enough, a contractive mapping $\boldsymbol{\Phi}_\delta : \mathscr{C}_{2,\delta}^p \to \mathscr{C}_{2,\delta}^p$ such that the unique fixed point $(\boldsymbol{\lambda}_t^\star)_{t \in [0,\delta]}$ is a weak solution of (S29). Considering $(\mathbf{W}_t^{\boldsymbol{\lambda}^\star})_{t \in [0,\delta]}$, we obtain the unique strong solution of (S29) on $[0, \delta]$.

Let $\delta > 0$. Denote $(\boldsymbol{\lambda}_t^{\boldsymbol{\mu}})_{t \in [0,\delta]} \in \mathscr{P}_2(\mathbb{R}^p)^{[0,\delta]}$ such that for any $t \in [0, \delta]$, $\boldsymbol{\lambda}_t^{\boldsymbol{\mu}}$ is the distribution of $\mathbf{W}_t^{\boldsymbol{\mu}}$ with initial condition $\mathbf{W}_0^\star$ with distribution $\boldsymbol{\lambda}_0^{\boldsymbol{\mu}} = \mu_0$. In addition, using (S1), (S27), (S30), **B**1, **B**2, the Cauchy-Schwarz inequality, the Itô isometry and the fact that for any $a, b \geq 0$, $2ab \leq a^2 + b^2$, there exists $C \geq 0$ such that for any $t, s \in [0, \delta]$ with $t \geq s$,

$$\mathcal{W}_2(\boldsymbol{\lambda}_t^{\boldsymbol{\mu}}, \boldsymbol{\lambda}_s^{\boldsymbol{\mu}})^2 \leq \mathbb{E}\left[\|\mathbf{W}_t^{\boldsymbol{\mu}} - \mathbf{W}_s^{\boldsymbol{\mu}}\|^2\right]$$

$$\leq 2\mathbb{E}\left[\left\|\int_s^t b(u, \mathbf{W}_u^{\boldsymbol{\mu}}, \boldsymbol{\mu}_u)\mathrm{d}u\right\|^2\right] + 2\mathbb{E}\left[\left\|\int_s^t \sigma(u, \mathbf{W}_u^{\boldsymbol{\mu}}, \boldsymbol{\mu}_u)\mathrm{d}\mathbf{B}_u\right\|^2\right]$$

$$\leq 2(t-s)\int_s^t \mathbb{E}\left[\|b(u, \mathbf{W}_u^{\boldsymbol{\mu}}, \boldsymbol{\mu}_u)\|^2\right]\mathrm{d}u + 2\int_s^t \mathbb{E}\left[\mathrm{Tr}(\sigma\sigma^\top(u, \mathbf{W}_u^{\boldsymbol{\mu}}, \boldsymbol{\mu}_u))\right]\mathrm{d}u$$

$$\leq 4(t-s)\int_s^t \left\{\|b(u, 0, \boldsymbol{\mu}_u)\|^2 + \mathtt{M}_1^2\mathbb{E}\left[\|\mathbf{W}_u^{\boldsymbol{\mu}}\|^2\right]\right\}\mathrm{d}u$$

$$+ 4 \int_s^t \left\{ \|\sigma(u, 0, \boldsymbol{\mu}_u)\|^2 + \mathtt{M}_1^2 \mathbb{E}\left[\|\mathbf{W}_u^{\boldsymbol{\mu}}\|^2\right] \right\} \mathrm{d}u$$

$$\leq 4(1+\delta)(t-s)\left[\mathtt{M}_1^2 \sup_{t\in[0,\delta]} \mathbb{E}[\|\mathbf{W}_t^{\boldsymbol{\mu}}\|^2] + \sup_{t\in[0,\delta]}\left\{\|b(t,0,\boldsymbol{\mu}_t)\|^2 + \|\sigma(t,0,\boldsymbol{\mu}_t)\|^2\right\}\right]$$

$$\leq C(t-s)\{1 + \sup_{t\in[0,\delta]} \mathbb{E}[\|\mathbf{W}_t^{\boldsymbol{\mu}}\|^2]\}\,.$$

Therefore, $(\boldsymbol{\lambda}_t^{\boldsymbol{\mu}})_{t\in[0,\delta]} \in \mathscr{C}_{2,\delta}^p$. Let $\boldsymbol{\Phi}_\delta : \mathscr{C}_{2,\delta}^p \to \mathscr{C}_{2,\delta}^p$ given for any $(\boldsymbol{\mu}_t)_{t\in[0,\delta]} \in \mathscr{C}_{2,\delta}^p$ by $\boldsymbol{\Phi}_\delta((\boldsymbol{\mu}_t)_{t\in[0,\delta]}) = (\boldsymbol{\lambda}_t^{\boldsymbol{\mu}})_{t\in[0,\delta]}$. Let $(\boldsymbol{\mu}_{1,t})_{t\in[0,\delta]}, (\boldsymbol{\mu}_{2,t})_{t\in[0,\delta]} \in \mathscr{C}_{2,\delta}^p$, using (S1), (S30), **B**1, **B**2, the Cauchy-Schwarz inequality, the Itô isometry, the fact that for any $a, b \geq 0$, $2ab \leq a^2 + b^2$ and Grönwall's inequality we have for any $t \in [0, \delta]$

$$\mathbb{E}\left[\|\mathbf{W}_t^{\boldsymbol{\mu}_1} - \mathbf{W}_t^{\boldsymbol{\mu}_2}\|^2\right] \leq 2\mathbb{E}\left[\left\|\int_0^t \{b(s, \mathbf{W}_s^{\boldsymbol{\mu}_1}, \boldsymbol{\mu}_{1,s}) - b(s, \mathbf{W}_s^{\boldsymbol{\mu}_2}, \boldsymbol{\mu}_{2,s})\} \mathrm{d}s\right\|^2\right]$$

$$+ 2\mathbb{E}\left[\left\|\int_0^t \{\sigma(s, \mathbf{W}_s^{\boldsymbol{\mu}_1}, \boldsymbol{\mu}_{1,s}) - \sigma(s, \mathbf{W}_s^{\boldsymbol{\mu}_2}, \boldsymbol{\mu}_{2,s})\} \mathrm{d}\mathbf{B}_s\right\|^2\right]$$

$$\leq 2\delta \int_0^t \mathbb{E}\left[\|b(s, \mathbf{W}_s^{\boldsymbol{\mu}_1}, \boldsymbol{\mu}_{1,s}) - b(s, \mathbf{W}_s^{\boldsymbol{\mu}_2}, \boldsymbol{\mu}_{2,s})\|^2\right] \mathrm{d}s$$

$$+ 2\int_0^t \mathbb{E}\left[\|\sigma(s, \mathbf{W}_s^{\boldsymbol{\mu}_1}, \boldsymbol{\mu}_{1,s}) - \sigma(s, \mathbf{W}_s^{\boldsymbol{\mu}_2}, \boldsymbol{\mu}_{2,s})\|^2\right] \mathrm{d}s$$

$$\leq 4\mathtt{M}_1^2(1+\delta) \int_0^t \left\{\mathbb{E}\left[\|\mathbf{W}_s^{\boldsymbol{\mu}_1} - \mathbf{W}_s^{\boldsymbol{\mu}_2}\|^2\right] + \int_{\mathsf{Z}} \zeta^2(z)\mathrm{d}\pi_{\mathsf{Z}}(z)\mathcal{W}_2^2(\boldsymbol{\mu}_{1,s}, \boldsymbol{\mu}_{2,s})\right\} \mathrm{d}s$$

$$\leq 4\mathtt{M}_1^2\delta(1+\delta) \int_{\mathsf{Z}} \zeta^2(z)\mathrm{d}\pi_{\mathsf{Z}}(z)\mathcal{W}_{2,\delta}^2(\boldsymbol{\mu}_1, \boldsymbol{\mu}_2) + 4\mathtt{M}_1^2(1+\delta) \int_0^t \mathbb{E}\left[\|\mathbf{W}_s^{\boldsymbol{\mu}_1} - \mathbf{W}_s^{\boldsymbol{\mu}_2}\|^2\right] \mathrm{d}s$$

$$\leq 4\mathtt{M}_1^2\delta(1+\delta) \exp\left[4\mathtt{M}_1^2(1+\delta)\delta \int_{\mathsf{Z}} \zeta^2(z)\mathrm{d}\pi_{\mathsf{Z}}(z)\right] \mathcal{W}_{2,\delta}^2(\boldsymbol{\mu}_1, \boldsymbol{\mu}_2)\,.$$

Using this result, we obtain that for any $(\boldsymbol{\mu}_{1,t})_{t\in[0,\delta]}, (\boldsymbol{\mu}_{2,t})_{t\in[0,\delta]} \in \mathrm{C}([0,\delta], \mathscr{P}_2(\mathbb{R}^p))$,

$$\mathcal{W}_{2,\delta}^2(\boldsymbol{\Phi}_\delta(\boldsymbol{\mu}_1), \boldsymbol{\Phi}_\delta(\boldsymbol{\mu}_2)) \leq \sup_{t\in[0,\delta]} \mathbb{E}\left[\|\mathbf{W}_t^{\boldsymbol{\mu}_1} - \mathbf{W}_t^{\boldsymbol{\mu}_2}\|^2\right]$$

$$\leq 4\mathtt{M}_1^2\delta(1+\delta) \exp\left[4\mathtt{M}_1^2(1+\delta)\delta \int_{\mathsf{Z}} \zeta^2(z)\mathrm{d}\pi_{\mathsf{Z}}(z)\right] \mathcal{W}_{2,\delta}^2(\boldsymbol{\mu}_1, \boldsymbol{\mu}_2)\,.$$

Hence, for $\delta > 0$ small enough, $\boldsymbol{\Phi}_\delta$ is contractive and since $\mathrm{C}([0,\delta], \mathscr{P}_2(\mathbb{R}^p))$ is a complete metric space, we get, using Picard fixed point theorem, that there exists a unique $(\boldsymbol{\lambda}_t^\star)_{t\in[0,\delta]} \in \mathrm{C}([0,\delta], \mathscr{P}_2(\mathbb{R}^p))$ such that, $\boldsymbol{\Phi}_\delta(\boldsymbol{\lambda}^\star) = \boldsymbol{\lambda}^\star$. For this $\boldsymbol{\lambda}^\star$, we have that $(\mathbf{W}_t^{\boldsymbol{\lambda}^\star})_{t\in[0,\delta]}$ is a strong solution to (S29). We have shown that (S29) admits a strong solution for any initial condition $\mu_0 \in \mathscr{P}_2(\mathbb{R}^p)$.

We now show that pathwise uniqueness holds for (S29). Let $(\mathbf{W}_t^1)_{t\in[0,\delta]}$ and $(\mathbf{W}_t^2)_{t\in[0,\delta]}$ be two strong solutions of (S29) such that $\mathbf{W}_0^1 = \mathbf{W}_0^2 = w_0 \in \mathbb{R}^p$. Let, $(\boldsymbol{\mu}_{1,t})_{t\in[0,\delta]}$ and $(\boldsymbol{\mu}_{2,t})_{t\in[0,\delta]}$ such that for any $t \in [0, \delta]$, $\boldsymbol{\mu}_{1,t}$ is the distribution of $\mathbf{W}_t^1$ and $\boldsymbol{\mu}_{2,t}$ the one of $\mathbf{W}_t^2$. Since $\boldsymbol{\Phi}_\delta$ admits a unique fixed point, we get that $\boldsymbol{\mu}_1 = \boldsymbol{\mu}_2$. Hence, $(\mathbf{W}_t^1)_{t\in[0,\delta]}$ and $(\mathbf{W}_t^2)_{t\in[0,\delta]}$ are strong solutions of (S30) with $\boldsymbol{\mu} \leftarrow \boldsymbol{\mu}_1 = \boldsymbol{\mu}_2$ and since pathwise uniqueness holds for (S30), we get that $(\mathbf{W}_t^1)_{t\in[0,\delta]} = (\mathbf{W}_t^1)_{t\in[0,\delta]}$.

$\square$

### S4.3 Main result

**Theorem S7.** *Assume* **B**1 *and* **B**2. *For any* $N \in \mathbb{N}^\star$, *let* $(\mathbf{W}_t^{1:N})_{t\geq 0}$ *be a strong solution of* (S26) *and for any* $N \in \mathbb{N}^\star$ *and* $k \in \{1, \ldots, N\}$, *let* $(\mathbf{W}_t^{k,\star})_{t\geq 0}$ *be a strong solution of* (S29) *with*

Brownian motion $(\mathbf{B}_t^k)_{t\geq 0}$. Assume that there exists $\mu_0 \in \mathscr{P}_2(\mathbb{R}^p)$ such that for any $N \in \mathbb{N}^\star$, $\mathbf{W}_0^{1:N} = \mathbf{W}_0^{\star,1:N}$ has distribution $\mu_0^{\otimes N}$. Then for any $T \geq 0$, $N \in \mathbb{N}^\star$ and $k \in \{1,\ldots,N\}$

$$\mathbb{E}\left[\sup_{t\in[0,T]} \left\|\mathbf{W}_t^{k,N} - \mathbf{W}_t^{k,\star}\right\|^2\right] \leq 32(1+T)^2\left(1 + \int_\mathsf{Z} \zeta^2(z)\mathrm{d}\pi_\mathsf{Z}(z)\right)\left(\mathtt{M}_2^2 N^{-2\kappa} + \mathtt{M}_1^2 N^{-1}\right)$$
$$\times \exp\left[16(1+T)^2\left(1 + \int_\mathsf{Z} \zeta^2(z)\mathrm{d}\pi_\mathsf{Z}(z)\right)\mathtt{M}_1^2\right].$$

*Proof.* Let $T \geq 0$. For any $N \in \mathbb{N}^\star$, $t \geq 0$, let $\boldsymbol{\nu}_t^{\star,N} = (1/N)\sum_{j=1}^N \delta_{\mathbf{W}_s^{\star,j}}$. Using **B1**, **B2**, Itô's isometry, Doob's inequality, Jensen's inequality and the fact that for any $a,b \geq 0$, $(a+b)^2 \leq 2(a^2 + b^2)$, we have for any $N \in \mathbb{N}^\star$ and $k \in \{1,\ldots,N\}$

$$\mathbb{E}\left[\sup_{t\in[0,T]} \left\|\mathbf{W}_t^{k,N} - \mathbf{W}_t^{k,\star}\right\|^2\right] \leq 2\mathbb{E}\left[\sup_{t\in[0,T]} \left\|\int_0^t \left(b_N(s, \mathbf{W}_s^{k,N}, \boldsymbol{\nu}_s^N) - b(s, \mathbf{W}_s^{k,\star}, \boldsymbol{\lambda}_s^\star)\right)\mathrm{d}s\right\|^2\right]$$

$$+ 2\mathbb{E}\left[\sup_{t\in[0,T]} \left\|\int_0^t \left(\sigma_N(s, \mathbf{W}_s^{k,N}, \boldsymbol{\nu}_s^N) - \sigma(s, \mathbf{W}_s^{k,\star}, \boldsymbol{\lambda}_s^\star)\right)\mathrm{d}\mathbf{B}_s^{k,N}\right\|^2\right]$$

$$\leq 2T\int_0^T \mathbb{E}\left[\left\|b_N(s, \mathbf{W}_s^{k,N}, \boldsymbol{\nu}_s^N) - b(s, \mathbf{W}_s^{k,\star}, \boldsymbol{\lambda}_s^\star)\right\|^2\right]\mathrm{d}s$$

$$+ 2\mathbb{E}\left[\left\|\int_0^T \left(\sigma_N(s, \mathbf{W}_s^{k,N}, \boldsymbol{\nu}_s^N) - \sigma(s, \mathbf{W}_s^{k,\star}, \boldsymbol{\lambda}_s^\star)\right)\mathrm{d}\mathbf{B}_s^{k,N}\right\|^2\right]$$

$$\leq 2(1+T)\int_0^T \left\{\mathbb{E}\left[\left\|b_N(s, \mathbf{W}_s^{k,N}, \boldsymbol{\nu}_s^N) - b(s, \mathbf{W}_s^{k,\star}, \boldsymbol{\lambda}_s^\star)\right\|^2\right]\right.$$

$$\left.+ \mathbb{E}\left[\left\|\sigma_N(s, \mathbf{W}_s^{k,N}, \boldsymbol{\nu}_s^N) - \sigma(s, \mathbf{W}_s^{k,\star}, \boldsymbol{\lambda}_s^\star)\right\|^2\right]\right\}\mathrm{d}s$$

$$\leq 8\mathtt{M}_2^2(1+T)^2 N^{-2\kappa} + 4(1+T)\int_0^T \left\{\mathbb{E}\left[\left\|b(s, \mathbf{W}_s^{k,N}, \boldsymbol{\nu}_s^N) - b(s, \mathbf{W}_s^{k,\star}, \boldsymbol{\lambda}_s^\star)\right\|^2\right]\right.$$

$$\left.+ \mathbb{E}\left[\left\|\sigma(s, \mathbf{W}_s^{k,N}, \boldsymbol{\nu}_s^N) - \sigma(s, \mathbf{W}_s^{k,\star}, \boldsymbol{\lambda}_s^\star)\right\|^2\right]\right\}\mathrm{d}s$$

$$\leq 8\mathtt{M}_2^2(1+T)^2 N^{-2\kappa} + 8\mathtt{M}_1^2(1+T)$$

$$\times \int_0^T \left\{\int_\mathsf{Z} \mathbb{E}\left[\left\|\boldsymbol{\nu}_s^N[\mathrm{g}(\cdot,z)] - \boldsymbol{\lambda}_s^\star[\mathrm{g}(\cdot,z)]\right\|^2\right]\mathrm{d}\pi_\mathsf{Z}(z) + \mathbb{E}\left[\left\|\mathbf{W}_s^{k,N} - \mathbf{W}_s^{k,\star}\right\|^2\right]\right\}\mathrm{d}s$$

$$\leq 8\mathtt{M}_2^2(1+T)^2 N^{-2\kappa} + 16\mathtt{M}_1^2(1+T)$$

$$\times \int_0^T \left\{\int_\mathsf{Z} \left(\mathbb{E}\left[\left\|\boldsymbol{\nu}_s^N[\mathrm{g}(\cdot,z)] - \boldsymbol{\nu}_s^{\star,N}[\mathrm{g}(\cdot,z)]\right\|^2\right] + \mathbb{E}\left[\left\|\boldsymbol{\nu}_s^{\star,N}[\mathrm{g}(\cdot,z)] - \boldsymbol{\lambda}_s^\star[\mathrm{g}(\cdot,z)]\right\|^2\right]\right)\mathrm{d}\pi_\mathsf{Z}(z)\right.$$

$$\left.+ \mathbb{E}\left[\left\|\mathbf{W}_s^{k,N} - \mathbf{W}_s^{k,\star}\right\|^2\right]\right\}\mathrm{d}s.$$

Then using the Cauchy-Schwarz's inequality, the fact that $\{(\mathbf{W}_t^{k,N})_{t\geq 0}\}_{k=1}^N$ are exchangeable, *i.e.* for any permutation $\tau : \{1,\ldots,N\} \to \{1,\ldots,N\}$, $\{(\mathbf{W}_t^{k,N})_{t\geq 0}\}_{k=1}^N$ has the same distribution as $\{(\mathbf{W}_t^{\tau(k),N})_{t\geq 0}\}_{k=1}^N$ and $\{(\mathbf{W}_t^{k,\star})_{t\geq 0}\}_{k=1}^N$ are independent we have

$$\mathbb{E}\left[\sup_{t\in[0,T]} \left\|\mathbf{W}_t^{k,N} - \mathbf{W}_t^{k,\star}\right\|^2\right] \leq 8\mathtt{M}_2^2(1+T)^2 N^{-2\kappa} + 16\mathtt{M}_1^2(1+T)$$

$$\times \int_0^T \left\{\frac{1}{N}\int_\mathsf{Z} \zeta^2(z)\mathrm{d}\pi_\mathsf{Z}(z)\sum_{j=1}^N \mathbb{E}\left[\left\|\mathbf{W}_s^{j,N} - \mathbf{W}_s^{j,\star}\right\|^2\right] + \mathbb{E}\left[\left\|\mathbf{W}_s^{k,N} - \mathbf{W}_s^{k,\star}\right\|^2\right]\right.$$

$$\left.+ \int_\mathsf{Z} \mathbb{E}\left[\left\|\frac{1}{N}\sum_{j=1}^N \mathrm{g}(\mathbf{W}_s^{j,\star},z) - \int_{\mathbb{R}^p} \mathrm{g}(\bar{w},z)\mathrm{d}\boldsymbol{\lambda}_s^\star(\bar{w})\right\|^2\right]\mathrm{d}\pi_\mathsf{Z}(z)\right\}\mathrm{d}s$$

$$\leq 8\mathtt{M}_2^2(1+T)^2N^{-2\kappa} + 16\mathtt{M}_1^2(1+T)\left(1+\int_{\mathsf{Z}}\zeta^2(z)\mathrm{d}\pi_{\mathsf{Z}}(z)\right)\int_0^T\mathbb{E}\left[\left\|\mathbf{W}_s^{k,N}-\mathbf{W}_s^{k,\star}\right\|^2\right]\mathrm{d}s$$

$$+ 16\mathtt{M}_1^2(1+T)N^{-1}\int_0^T\int_{\mathsf{Z}}\mathbb{E}\left[\left\|\mathrm{g}(\mathbf{W}_s^{k,\star},z)-\int_{\mathbb{R}^p}\mathrm{g}(\bar{w},z)\mathrm{d}\boldsymbol{\lambda}_s^\star(\bar{w})\right\|^2\right]\mathrm{d}\pi_{\mathsf{Z}}(z)\mathrm{d}s$$

$$\leq 8\mathtt{M}_2^2(1+T)^2N^{-2\kappa} + 16\mathtt{M}_1^2(1+T)\left(1+\int_{\mathsf{Z}}\zeta^2(z)\mathrm{d}\pi_{\mathsf{Z}}(z)\right)\int_0^T\mathbb{E}\left[\left\|\mathbf{W}_s^{k,N}-\mathbf{W}_s^{k,\star}\right\|^2\right]\mathrm{d}s$$

$$+ 32\mathtt{M}_1^2(1+T)^2N^{-1}\left(1+\int_{\mathsf{Z}}\zeta^2(z)\mathrm{d}\pi_{\mathsf{Z}}(z)\right)\ .$$

We conclude the proof upon combining this result and Grönwall's inequality. $\qquad\square$

### S4.4 Proofs of the main results

In this section we prove Theorem 1, Theorem 2, Theorem S2, Theorem S3. Note that we only need to show Theorem S2 and Theorem S3, since in the case $\eta = 0$, Theorem S2 boils down to Theorem 1 and Theorem S3 to Theorem 2.

*Proof of Theorem S2.* Define for any $N \in \mathbb{N}^\star$, $w \in \mathbb{R}^p$, $\mu \in \mathscr{P}_2(\mathbb{R}^p)$ and $t \geq 0$

$$b_N(t,w,\mu) = (t+1)^{-\alpha}h(w,\mu)\ ,\ \sigma_N(t,w,\mu) = (t+1)^{-\alpha}((\gamma_{\alpha,\beta}(N)/M)^{1/2}\Sigma^{1/2}(w,\mu),\sqrt{2}\,\mathrm{Id})\ ,$$
$$b(t,w,\mu) = (t+1)^{-\alpha}h(w,\mu)\ ,\quad \sigma(t,w,\mu) = (t+1)^{-\alpha}(0,\sqrt{2}\,\mathrm{Id})\ ,$$

with $h$ and $\Sigma$ given in (S7). Using Proposition S4, we get that **B1** holds with $\mathtt{M}_1 \leftarrow \mathtt{L}$ and $\gamma_{\alpha,\beta}(N) = \gamma^{1/(1-\alpha)}N^{(\beta-1)/(1-\alpha)}$. In addition, using Proposition S4, **B2** holds with $\mathtt{M}_2 \leftarrow (\gamma^{1-\alpha}/M)^{1/2}p\mathtt{L}$ and $2\kappa = (1-\beta)/(1-\alpha)$. We conclude using Theorem S7. $\qquad\square$

*Proof of Theorem S3.* Define for any $N \in \mathbb{N}^\star$, $w \in \mathbb{R}^p$, $\mu \in \mathscr{P}_2(\mathbb{R}^p)$ and $t \geq 0$

$$b_N(t,w,\mu) = (t+1)^{-\alpha}h(w,\mu)\ ,\ \sigma_N(t,w,\mu) = (t+1)^{-\alpha}((\gamma^{1/(1-\alpha)}/M)^{1/2}\Sigma^{1/2}(w,\mu),\sqrt{2}\,\mathrm{Id})\ ,$$

with $h$ and $\Sigma$ given in (S7). Using Proposition S4, we get that **B1** holds with $\mathtt{M}_1 \leftarrow \mathtt{L}$. In addition, **B2** holds with $b = b_N$, $\sigma = \sigma_N$, $\mathtt{M}_2 \leftarrow 0$ and $\kappa = 0$. We conclude using Theorem S7. $\qquad\square$

*Proof of Proposition 4.* We consider only the case $\beta = 1$, the proof for $\beta \in [0,1)$ following the same lines. Let $M \in \mathbb{N}^\star$. We have for any $N \in \mathbb{N}^\star$ using Proposition S1,

$$\mathcal{W}_2(\boldsymbol{\Upsilon}^N,\delta_{\boldsymbol{\lambda}^\star})^2 \leq \mathbb{E}\left[\mathcal{W}_2(\boldsymbol{\nu}^N,\boldsymbol{\lambda}^\star)^2\right]$$
$$\leq N^{-1}\sum_{k=1}^N\mathbb{E}\left[\mathcal{W}_2(\delta_{(\mathbf{W}_t^{k,N})_{t\geq 0}},\boldsymbol{\lambda}^\star)^2\right] \leq N^{-1}\sum_{k=1}^N\mathbb{E}\left[\mathrm{m}^2((\mathbf{W}_t^{k,N})_{t\geq 0},(\mathbf{W}_t^{k,\star})_{t\geq 0})\right]\ . \quad\text{(S31)}$$

Let $\varepsilon > 0$ and $n_0$ such that $\sum_{n=n_0+1}^{+\infty}2^{-n} \leq \varepsilon$. Combining (S31), Theorem 1 and the Cauchy-Schwarz inequality we get that for any $N \in \mathbb{N}^\star$

$$\mathcal{W}_2(\boldsymbol{\Upsilon}^N,\delta_{\boldsymbol{\lambda}^\star})^2 \leq 2\varepsilon^2 + \frac{2n_0}{N}\sum_{k=1}^N\sum_{n=1}^{n_0}\mathbb{E}\left[\sup_{t\in[0,n]}\|\mathbf{W}_t^{k,N}-\mathbf{W}_t^{k,\star}\|^2\right] \leq 2\varepsilon^2 + 2n_0N^{-1}\sum_{n=0}^{n_0}C_{1,n}\ .$$

Therefore, for any $\varepsilon > 0$ there exists $N_0 \in \mathbb{N}^\star$ such that for any $N \in \mathbb{N}^\star$ with $N \geq N_0$, $\mathcal{W}_2(\boldsymbol{\Upsilon}^N,\delta_{\boldsymbol{\lambda}^\star}) \leq \varepsilon$, which concludes the proof. $\qquad\square$

## S5 Existence of invariant measure in the one-dimensional case

In this section we prove Proposition 5.

*Proof of Proposition 5.* Since $V$ is $\eta$-strongly convex it admits a unique minimum at $w_0 \in \mathbb{R}$. Using **A**1-(c), the fact that $V$ is $\eta$-strongly convex and [9, Theorem 2.1.5, Theorem 2.1.7] there exists $\mathtt{M} \geq 0$ such that for any $w \in \mathbb{R}$ we have

$$\eta(w-w_0)^2/2 \leq V(w) - V(w_0) \leq \mathtt{M}(w-w_0)^2/2\ . \quad\text{(S32)}$$

In addition, using Proposition S4, we have for any $\mu \in \mathscr{P}_2(\mathbb{R})$ and $w \in \mathbb{R}$,

$$\bar{\sigma}^2 \leq \Sigma(w, \mu) \leq \mathtt{L}^2 \ . \tag{S33}$$

Recall that for any $\mu \in \mathscr{P}_2(\mathbb{R})$ and $w \in \mathbb{R}$, $h(w, \mu) = \bar{h}(w, \mu) + V'(w)$, with $\bar{h}$ given in (S7). Note that for any $w \in [w_0, +\infty)$, $V'(w) \geq 0$ and for any $w \in (-\infty, w_0]$, $V'(w) \leq 0$. Combining this result, Proposition S4, (S32) and (S33), there exists $\mathtt{m}_1 > 0$ and $c_1 \in \mathbb{R}$ such that for any $\mu \in \mathscr{P}_2(\mathbb{R})$ and $w \in \mathbb{R}$, we have distinguishing the case $w \leq w_0$ and $w > w_0$,

$$
\begin{aligned}
\int_0^w \{h/\Sigma\}(\tilde{w}, \mu)\mathrm{d}w &\geq -\bar{\sigma}^{-2}\mathtt{L}^2\,|w| + \int_0^w V'(\tilde{w})/\Sigma(\tilde{w}, \mu)\mathrm{d}\tilde{w} \\
&\geq -\bar{\sigma}^{-2}\mathtt{L}^2\,|w| - \bar{\sigma}^{-2}\sup_{\tilde{w} \in [0, w_0]}|V'(\tilde{w})|\,|w_0| + \int_{w_0}^w V'(\tilde{w})/\Sigma(\tilde{w}, \mu)\mathrm{d}\tilde{w} \\
&\geq -\bar{\sigma}^{-2}\mathtt{L}^2\,|w| - \bar{\sigma}^{-2}\sup_{\tilde{w} \in [0, w_0]}|V'(\tilde{w})|\,|w_0| + (V(w) - V(w_0))\mathtt{L}^{-2} \geq \mathtt{m}_1 w^2 + c_1 \ .
\end{aligned}
\tag{S34}
$$

Therefore, we obtain that for any $\mu \in \mathscr{P}_2(\mathbb{R})$, $\int_{\mathbb{R}} \exp[\int_0^w h(\tilde{w}, \mu)/\Sigma(\tilde{w}, \mu)\mathrm{d}\tilde{w}]\mathrm{d}w < +\infty$. Define $H : \mathscr{P}_2(\mathbb{R}) \to \mathscr{P}_2(\mathbb{R})$ such that for any $\mu \in \mathscr{P}_2(\mathbb{R})$, $H(\mu)$ is the probability measure with density $\rho_\mu$ given for any $w \in \mathbb{R}$ by

$$\rho_\mu(w) \propto \bar{\Sigma}^{-1}(w, \mu)\exp\left[-2\int_0^w h(\tilde{w}, \mu)/\bar{\Sigma}(\tilde{w}, \mu)\mathrm{d}\tilde{w}\right] \ ,$$

where $\bar{\Sigma}(w, \mu) = \gamma^{1/(1-\alpha)}\Sigma(w, \mu)/M$. Similarly to (S34), there exist $\mathtt{m}_2 > 0$ and $c_2 \in \mathbb{R}$ such that for any $\mu \in \mathscr{P}_2(\mathbb{R})$ and $w \in \mathbb{R}$

$$\int_0^w h(\tilde{w}, \mu)/\Sigma(\tilde{w}, \mu)\mathrm{d}\tilde{w} \leq \mathtt{m}_2 w^2 + c_2 \ . \tag{S35}$$

Combining (S33), (S34) and (S35), there exists $\mathtt{m} > 0$ and $c \in \mathbb{R}$ such that for any $\mu \in \mathscr{P}_2(\mathbb{R})$ and $w \in \mathbb{R}$, $\rho_\mu(w) \leq c\mathrm{e}^{-\mathtt{m}w^2}$. Using this result, we get that $\sup_{\mu \in \mathscr{P}_2(\mathbb{R})} \int_{\mathbb{R}} w^4 \rho_\mu(w)\mathrm{d}w < +\infty$. Therefore, using [10, Theorem 2.7] we obtain that $H(\mathscr{P}_2(\mathbb{R}))$ is relatively compact in $(\mathscr{P}_2(\mathbb{R}), \mathcal{W}_2)$.

We now show that $H \in \mathrm{C}(\mathscr{P}_2(\mathbb{R}), \mathscr{P}_2(\mathbb{R}))$. Let $\mu \in \mathscr{P}_2(\mathbb{R})$ and $(\mu_n)_{n \in \mathbb{N}} \in \mathscr{P}_2(\mathbb{R})^{\mathbb{N}}$ such that $\lim_{n \to +\infty} \mu_n = \mu$. Using Proposition S4 and the Lebesgue dominated convergence theorem we obtain that for any $w \in \mathbb{R}$, $\lim_{n \to +\infty} \rho_{\mu_n}(w) = \rho_\mu(w)$. Using Scheffé's lemma we get that $\lim_{n \to +\infty} \int_{\mathbb{R}} |\rho_{\mu_n}(w) - \rho_\mu(w)|\,\mathrm{d}w = 0$. Hence, $(H(\mu_n))_{n \in \mathbb{N}}$ weakly converges towards $H(\mu)$.

Let $(H(\mu_{n_k}))_{k \in \mathbb{N}}$ be a converging sequence in $(\mathscr{P}_2(\mathbb{R}), \mathcal{W}_2)$. Therefore, $(H(\mu_{n_k}))_{k \in \mathbb{N}}$ also weakly converges and we obtain that $\lim_{k \to +\infty} \mathcal{W}_2(H(\mu_{n_k}), H(\mu)) = 0$. Since $\{H(\mu_n) : n \in \mathbb{N}\}$ is relatively compact and admits a unique limit point we obtain that $\lim_{n \to +\infty} \mathcal{W}_2(H(\mu_n), H(\mu)) = 0$.

Hence $H \in \mathrm{C}(\mathscr{P}_2(\mathbb{R}), \mathscr{P}_2(\mathbb{R}))$. Therefore, since $H \in \mathrm{C}(\mathscr{P}_2(\mathbb{R}), \mathscr{P}_2(\mathbb{R}))$ and $H(\mathscr{P}_2(\mathbb{R}))$ is relatively compact in $\mathscr{P}_2(\mathbb{R})$ Schauder's theorem [11, Appendix] implies that $H$ admits a fixed point.

Let $\mu \in \mathscr{P}_2(\mathbb{R})$ be a fixed point of $H$. We now show that $\mu$ is an invariant probability distribution for (8). Let $(\mathbf{W}_t^\mu)_{t \geq 0}$ such that $\mathbf{W}_0^\mu$ has distribution $\mu$ and strong solution to the following SDE

$$\mathrm{d}\mathbf{W}_t^\mu = h(t, \mu)\mathrm{d}t + \gamma^{1/(1-\alpha)}\Sigma(\mathbf{W}_t^\mu, \mu)\mathrm{d}\mathbf{B}_t \ . \tag{S36}$$

An invariant distribution for (S36) is given by $H(\mu)$, see [12]. Hence, since $\mu = H(\mu)$, for any $t \geq 0$, $\mathbf{W}_t^\mu$ has distribution $\mu$ and $(\mathbf{W}_t^\mu)_{t \geq 0}$ is a strong solution to (8). Therefore, $\mu$ is an invariant probability measure for (8) which concludes the proof. $\qquad\square$

## S6 Links with gradient flow approach

**Case $\beta \in [0, 1)$** We now focus on the mean-field distribution $\boldsymbol{\lambda}^\star$. Note that the trajectories of $(\mathbf{W}_t^{k,\star})_{t \geq 0}$ for any $k \in \mathbb{N}^\star$ are deterministic conditionally to $\mathbf{W}_0^{k,\star}$. Using Itô's formula, we obtain that for any function $f \in \mathrm{C}^2(\mathbb{R}^p)$ with compact support and $t \geq 0$

$$\int_{\mathbb{R}^p} f(\tilde{w})\mathrm{d}\boldsymbol{\lambda}_t^\star(\tilde{w}) = \int_{\mathbb{R}^p} f(\tilde{w})\mathrm{d}\mu_0(\tilde{w}) + \int_0^t \int_{\mathbb{R}^p} (s+1)^{-\alpha}\langle h(\tilde{w}, \boldsymbol{\lambda}_s^\star), \nabla f(\tilde{w})\rangle\mathrm{d}\boldsymbol{\lambda}_s^\star(\tilde{w}) \ . \tag{S37}$$

Therefore, if for any $t \geq 0$, $\boldsymbol{\lambda}_t^\star$ admits a density $\boldsymbol{\rho}_t^\star$ such that $(\boldsymbol{\rho}_t^\star)_{t\geq0} \in \mathrm{C}^1(\mathbb{R}_+ \times \mathbb{R}^p, \mathbb{R})$ we obtain that $(\boldsymbol{\rho}_t)_{t\geq0}$ satisfies the following evolution equation for any $t > 0$ and $w \in \mathbb{R}^p$

$$\partial_t \boldsymbol{\rho}_t^\star(w) = -(t+1)^{-\alpha}\mathrm{div}(\bar{h}(\cdot, \boldsymbol{\rho}_t^\star)\boldsymbol{\rho}_t^\star)(w) \, ,$$

with for any $w \in \mathbb{R}^p$ and $\mu \in \mathscr{P}(\mathbb{R}^p)$ with density $\rho$, $h(w, \mu) = \bar{h}(w, \rho)$. In the case $\alpha = 0$, it is well-known, see [13, 4, 14], that $(\boldsymbol{\rho}_t^\star)_{t\geq0}$ is a Wasserstein gradient flow for the functional $\mathscr{R}^\star : \mathscr{P}_2^c(\mathbb{R}^p) \to \mathbb{R}$ given for any $\rho \in \mathscr{P}_2^c(\mathbb{R}^p)$

$$\mathscr{R}^\star(\rho) = \int_{\mathsf{X}\times\mathsf{Y}} \ell\left(\int_{\mathbb{R}^p} F(\tilde{w}, x)\rho(\tilde{w})\mathrm{d}\tilde{w}, y\right) \mathrm{d}\pi(x, y) \, , \tag{S38}$$

where $\mathscr{P}_2^c(\mathbb{R}^p)$ is the set of probability density satisfying $\int_{\mathbb{R}^p} \|\tilde{w}\|^2 \rho(\tilde{w})\mathrm{d}\tilde{w} < +\infty$.

**Case $\beta = 1$**  Focusing on $(\boldsymbol{\lambda}_t^\star)_{t\geq0}$, we no longer obtain that $(\boldsymbol{\lambda}_t^\star)_{t\geq0}$ is a gradient flow for (S38). Indeed, using Itô's formula, we have the following evolution equation for any $f \in \mathrm{C}_c^2(\mathbb{R}^p)$ and $t \geq 0$

$$\int_{\mathbb{R}^p} f(\tilde{w})\mathrm{d}\boldsymbol{\lambda}_t^\star(\tilde{w}) = \int_{\mathbb{R}^p} f(\tilde{w})\mathrm{d}\mu_0(\tilde{w}) + \int_0^t \int_{\mathbb{R}^p} (s+1)^{-\alpha}\langle h(\tilde{w}, \boldsymbol{\lambda}_s^\star), \nabla f(\tilde{w})\rangle \mathrm{d}\boldsymbol{\lambda}_s^\star(\tilde{w})$$
$$+ \int_0^t \int_{\mathbb{R}^p} (s+1)^{-\alpha} \mathrm{Tr}(\Sigma(\tilde{w}, \boldsymbol{\lambda}_s^\star)\nabla^2 f(\tilde{w}))\mathrm{d}\tilde{w} \, . \tag{S39}$$

We higlight that the additional term in (S39) from (S37) corresponds to some entropic regularization of the risk $\mathscr{R}^\star$. Indeed, if for any $w \in \mathbb{R}^p$ and $\mu \in \mathscr{P}(\mathbb{R}^p)$, $\Sigma = \beta \, \mathrm{Id}$ then, in the case $\alpha = 0$, we obtain that $(\boldsymbol{\rho}_t^\star)_{t\geq0}$ is a gradient flow for $\rho \mapsto U^\star(\rho) + \beta \mathrm{Ent}(\rho)$, where $\mathrm{Ent} : \mathsf{K}_2 \to \mathbb{R}$ is given for any $\rho \in \mathsf{K}_2$ by

$$\mathrm{Ent}(\rho) = -\int_{\mathbb{R}^p} \rho(x) \log(\rho(x))\mathrm{d}x \, .$$

This second regime emphasizes that large stepsizes act as an implicit regularization procedure for SGD.

## S7  Additional Experiments

In this section we present additional experiments illustrating the convergence results of the empirical measures. Contrary to the main document we illustrate our results with histograms of the weights of the first and second layers of the network, with a large number of different values of the parameters $\alpha$, $\beta$ and $N$.

**Setting.**  In order to perform the following experiments we implemented a two-layer fully connected neural network on PyTorch. The input layer has the size of the input data, *i.e.*, $N_{\mathrm{input}} = 28 \times 28$ units in the case of the MNIST dataset [15] and $N_{\mathrm{input}} = 32 \times 32 \times 3$ in the case of the CIFAR-10 dataset [16]. We use a varying number of $N$ units in the hidden layer and the output layer has 10 units corresponding to the 10 possible labels of the classification tasks. We use a ReLU activation function and the cross-entropy loss.

The linear layers' weights are initialized with PyTorch default initialization function which is a uniform initialization between $-1/N_{\mathrm{input}}^{1/2}$ and $1/N_{\mathrm{input}}^{1/2}$. In all our experiments, if not specified, we consider an initialization $\mathbf{W}_0^{1:N}$ with distribution $\mu_0^{\otimes N}$ where $\mu_0$ is the uniform distribution on $[-0.04, 0.04]$.

In order to train the network we use SGD as described in Section 2 with an initial learning rate of $\gamma N^\beta$. In the case where $\alpha > 0$ we decrease this stepsize at each iteration to have a learning rate of $\gamma N^\beta(n + \gamma_{\alpha,\beta}(N)^{-1})^{-\alpha}$. All experiments on the MNIST dataset are run for a finite time horizon $T = 100$ and the ones on the CIFAR-10 dataset are run for $T = 10000$. The average runtime of the experiments for $N = 50000$ on the MNIST dataset is one day and the experiments on the CIFAR-10 dataset run during two days. The experiments were run on a cluster of 24 CPUs with 126Go of RAM.

All the histograms represented below correspond to the first coordinate of the weights' vector.

**Experiments.** Figure S1 shows that the empirical distributions of the weights converge as the number of hidden units $N$ goes to infinity. Those figures illustrate also the fact that we obtain two different limiting distributions one for $\beta < 1$ (represented on the 3 first figures) and one for $\beta = 1$ (on the last figure). The results presented on Figure S2 illustrate the same fact, one the second layer. This means that the results we stated in Section 3 are also true for the weights of the second layer, thanks to the procedure described for example in [13].

Figure S1: Convergence of the weights of the first layer as $N \to +\infty$ for $\alpha = 0$ and $M = 100$. The first line corresponds to $\beta = 0.25$, the second to $\beta = 0.5$, the third to $\beta = 0.75$ and the last line to $\beta = 1.0$.

Figure S2: Convergence of the weights of the first layer as $N \to +\infty$ for $\alpha = 0$ and $M = 100$. The first line corresponds to $\beta = 0.25$, the second to $\beta = 0.5$, the third to $\beta = 0.75$ and the last line to $\beta = 1.0$.

On Figure S3 and Figure S4 we show the results of the exact same experiments but this time using decreasing stepsizes and a parameter $\alpha = 0.25$. Once again our experiments illustrate the convergence of the empirical distributions to some limiting distribution, and we can also identify two regimes. Note that the limiting distribution satisfying (S37) or (S39) (depending on the value of $\beta$), it depends on the parameter $\alpha$. Therefore the limiting distribution obtained in the case where $\alpha = 0.25$ is different from the one obtained when $\alpha = 0$. This is particularly visible in the case where $\beta = 1$ (as shown in green on Figure S3 and Figure S4).

Figure S3: Convergence of the weights of the first layer as $N \to +\infty$ for $\alpha = 0.25$ and $M = 100$. The first line corresponds to $\beta = 0.5$, the second to $\beta = 0.75$ and the last line to $\beta = 1.0$.

Figure S4: Convergence of the weights of the first layer as $N \to +\infty$ for $\alpha = 0.25$ and $M = 100$. The first line corresponds to $\beta = 0.5$, the second to $\beta = 0.75$ and the last line to $\beta = 1.0$.

We now study the role of the batch size $M$ on the convergence toward the mean-field regime. Figure S5 illustrates the convergence of the empirical measures in the case where $\beta < 1$ (here $\beta = 0.75$) of the weights of the hidden layer of the neural network, for a fixed number of neurons $N = 10000$ for different batch sizes $M$. We indeed observe convergence with $M$.

Figure S5: Convergence of the weights as $M \to \infty$