[Reviews · NeurIPS 2020]

Review 1

Summary and Contributions: This article considers the infinite width limit of SGD for one layer networks in which only the first layer is trained and the learning rate is allowed to depend on the number of hidden units. The main result is the identification of two different limiting behaviors for the weight dynamics. In the first regime, when the learning rate is below a critical threshold, the limiting dynamics of particles (weights) is a deterministic ODE in which weights for different neurons are independent. In the second regime, in contrast, the limiting dynamics of the weights is an SDE in which weights corresponding to different neurons are again independent. Some numerical results comparing training on MNIST and CIFAR10 show excellent agreement between the statistics of weights learned by a wide network and the corresponding theoretical predictions.

Strengths: The question of the effect of large learning rate is interesting both practically and theoretically. This paper obtains a strong result on this subject. Not only it is shown that there is a qualitative (and quantitative) difference between large and small learning rate, but this article in fact obtains a formula for the limiting dynamics and interprets it as a discretization of McKean-Vlasov on empirical measures of weights.

Weaknesses: The main weakness of this article is the restricted setting in which it applies. Namely, the authors consider training just the first layer weights (with output weights set to 1/#neurons) in a one layer network. This is a reasonable starting case but already having networks with more than one layer is very interesting due to the fact that neurons are no longer independent.

Correctness: I did not check every detail of the derivations, but the method seems correct.

Clarity: For the most part, yes, but there are a couple of specific points that I think should be clarified (see below).

Relation to Prior Work: yes

Reproducibility: Yes

Additional Feedback: I have a few minor comments. (1) The overall discussion of the learning rate is somewhat confusing and could be clarified. Specifically: (1a) Depending on how one thinks about it, the learning rate in previous papers on infinite width SGD depends on the number of hidden units. What I mean is that you explicitly put in the 1/N as the size of the weights into the last layer. This has the effect of putting in a 1/N in the derivative d Loss / d W, where W is a weight in the first layer, which is akin to putting an extra 1/N into the learning rate. (I understand you put in the 1/N into the definition since you wanted a well-defined limit.) In previous papers (NTK-type analyses in deeper networks), sometimes this scale of weights is like N^{-1/2}. (1b) The form of the learning rate \gamma N^\beta ( n + \gamma_{\alpha\beta}(N)^{-1} )^{-\alpha} is hard to intuitively understand. Indeed, while I understand that this is what you need to get a mean field limit, the factor of N^\beta out in from seems confusing and is only properly understood I think when you realize that the 1/N from the weights in the 2nd layer actually turns it into a N^{\beta-1} (this also clarifies why \beta=1 is special). Also, what is the role of the n? This tells me that, unless \alpha=0, the step size decays as a function of time at a very specific rate. Is that important/interesting? (2) After corollary 3, you say that the result is valid for the whole trajectory and not just for a fixed time horizon. This is confusing since the constants C_{m,T} in Theorems 1 and 2 depend on T. It would be nice if you explained a bit more here what is meant. ____________________________________ POST REBUTTAL UPDATE: I have read the authors' feedback. I am glad that the authors will put in an extended discussion of how to interpret the learning rate in terms of alpha, beta, N, n. I think this will make the paper more readable. My overall assessment of the article remains positive (7/10).


Review 2

Summary and Contributions: This paper performs a mean-field analysis of the behavior of SGD on a 1-hidden layer neural network with number of neurons N. One question is how the step-size-dependence-on-N affects the behavior as N grows, both in the case of a step size that is fixed over iterations, and in the case where the step size decays as a power law over time. Roughly speaking, there is a step-size-dependence-on-N like $N^\beta$ that emerges from the analysis, for which the $\beta \in [0, 1)$ regime behaves differently from the $\beta = 1$ regime. The paper runs some experiments on MNIST and CIFAR to show 1-layer networks converging in distribution as N grows.

Strengths: The main novelty of this work is in studying the $\beta > 0$ case, the $\beta = 0$ case has been studied before. I think the regime change result is interesting: It suggests that one can choose a larger step size for wide networks as long as the step-size is sublinear in the width.

Weaknesses: The theoretical work requires pretty strict assumptions in the form of A1-(a-d), in particular A1-b rules out non-smooth activation functions, as well as functions that are unbounded for fixed x. Perhaps one way in which the results could be more thorough is to cover what happens in cases where the step size is still proportional to N^{\beta} at high iteration number, but not with the exact $\gamma_{\alpha, \beta)(N)$ additive factor.

Correctness: The experiments use the ReLU, yet your assumptions assume the non-linearity is thrice-differentiable. By and large though, the approach seems correct.

Clarity: I find the paper clearly written.

Relation to Prior Work: Yes, the work describes several prior works that deal with subcases of the more general framework it puts forward.

Reproducibility: Yes

Additional Feedback: Some things that seemed like typos to me: Line 91: $\ell$ is a function of $\mathbb{R} \times \mathbb{R}$, I think it should be $\mathbb{R} \times \mathsf{Y}$? Line 110: missing period. Line 207: Should be "(7) if $\beta \in [0,1)$"? EDIT 8/22/2020 The author response says the learning rate will be generalized from $(n + \gamma_{\alpha, \beta} (N) N^{-1})^\alpha$ to $(n + c)^\alpha$. I think this really helps the usability of the result of this paper in further work, and it pushes the paper up from marginal accept to an accept for me.


Review 3

Summary and Contributions: This paper studies SGD dynamics for two-layers neural networks approximating it by a mean field diffusion in the large width limit. In the mean field diffusion, each neuron evolves independently according to a diffusion whose coefficients are functions of the overall density. The main results are bound on the distance between the original SGD dynamics and the mean field model. The authors study a family of scalings of the stepsize and show that depending on the scaling, the mean field diffusion is deterministic or not.

Strengths: 1) This work identify a new regime in which the mean field dynamics of each neuron is non-deterministic. This is an interesting phenomenon. 2) The mathematical analysis is quite sophisticated.

Weaknesses: 1) In the simpler deterministic regime of earlier work (corresponding to small stepsize), this paper does not seem to provide stronger results or new insights. 2) It is unclear what insight emerges from this new analysis that was not already in previous work. 3) It is unclear what is the difference between the two regimes from a machine learning point of view. Simulations point at a difference in generalization error, but it is unclear whether the theory sheds any light on this difference. Also, it is unclear whether this difference is generic.

Correctness: Both the claims and approach seem reasonable. I did not check the proof details.

Clarity: The paper is quite clear.

Relation to Prior Work: The comparison with earlier work seem to imply that the novelty of the new results id that they are quantitative, while pervious work only established asymptotics. This is wrong. Several earlier papers obtained quantitative bounds very similar to the current one (for instance [23,25]). The real novelty is in identifying a new non-deterministic limit.

Reproducibility: Yes

Additional Feedback:


Review 4

Summary and Contributions: The paper considers letting the gradient of two layered over-parametrized modela, with the second layer having fixed weights, vary with the number of neurons N. The paper shows that the choice of gradient scaling with N leads to two different mean-field behavior: a deterministic regime that has already been studied a new stochastic one that takes the form of a McKean-Vlasov diffusion.

Strengths: The paper’s claims are particularly elegant in that they derive from a “simple” modification of the gradient scaling as a function of the number of neurons N. The claims relating to the limiting behavior as N->\infty are supported by extensive theoretical results as well as empirical studies using real datasets.

Weaknesses: None worth discussing. The applicability of the results may be limited at this point, but the paper is likely an important stepping stone towards a better characterization of the generalization properties of neural networks.

Correctness: I am unable to fully ascertain the correctness of the results given the extensive supplementary materials needed to support them. However, I have found no obvious flaws, and the empirical studies do support the theoretical results.

Clarity: This is a very theoretical paper which dense in notation and with results left to the supplementary materials - not the easiest material to present concisely without losing the reader. To the extent that it can be done, I would argue this paper hits the mark. That being said, I do wonder about the assumptions in A1 and whether they belong in the paper rather than the supplementary materials. They are presented without much explanation, and from then on, they mostly serve as references in the theorems and propositions. An exposition of their significance and why they are needed, with the details left to the supplementary materials, might have been a more effective use of space.

Relation to Prior Work: The paper fits in a much larger context of understanding gradient descent for overparametrized models and the authors take the time to provide a clear exposition of where this particular contribution fits and in what sense it is an extension of previous work.

Reproducibility: Yes

Additional Feedback: Very interesting results. The use of only one layer with trainable weights makes sense from a theoretical perspective but it might still be interesting to find out what happens in practice if the parameters of both layers can be learned. Is that something you’ve explored? Do the one layer results represent a decent approximation or is a completely different dynamic at play in that case? I do realize this would amount to pure numerical studies, which is not the point of this paper, but I can’t help but wonder how different the two cases in the over-parametrized limit.

[Author Response · NeurIPS 2020]



*We thank all the four reviewers for their constructive and positive feedback. Please find our answers to major questions raised. Other points will be dealt with in the revised version.*

**Limited setting (Reviewers #1 and #4).** We acknowledge that our current setting is limited (two-layer neural networks with fixed second layer) and that considering more general architectures is a natural follow up of our work. Preliminary results indicate that our approach can be extended to the full two-layer neural network setting. However, note that the derivation of these results is based on the submitted paper. We think that a manuscript gathering the two settings would be too long and that is why we decided to first consider the simpler case and leave the extensions for future work. In addition, we emphasize that the two-layer neural network setting is common when studying the effect of overparameterization in neural networks. Finally, note that even though this assumption is restrictive, recent works (Suzuki [2020]) have pointed out that ResNets can be rigorously approached by a sum of two-layer neural networks.

**Relevance of the parameter $\alpha$ (Reviewer #1).** First recall that $\alpha$ corresponds to the decay power of the stepsize. Our motivation to consider decreasing stepsizes comes from the fact that they are commonly used in practice. Besides, another reason follows from a higher order analysis that we briefly explain. Whereas the convergence rates we derive only depend on $\beta \in [0,1]$, if we now turn to a Central Limit Theorem (CLT), as established in Sirignano and Spiliopoulos [2020] for $\beta = \alpha = 0$, preliminary computations suggest that we obtain convergence rates of the form $N^{(1-\beta)/(2-2\alpha)}$. Hence, we expect an interplay between $\alpha$ and $\beta$ to appear in this weak expansion. The rigorous derivation of such results is left for future work. We will add a discussion on this matter.

**Form of the learning rate (Reviewers #1 and #2).** In the revised version of our manuscript, we will explain in more details the dependency with respect to $\beta$. More precisely we will highlight that even though the learning rate in SGD scales as $\mathcal{O}(N^\beta)$ the learning rate in the mean-field dynamics scales as $\mathcal{O}(N^{\beta-1})$. In addition, we will emphasize that the term $(n + \gamma_{\alpha,\beta}(N)(N)^{-1})^{-\alpha}$ can be replaced by $(n+c)^{-\alpha}$ with $c > 0$ at the cost of modifying the limiting SDE.

**Comparison with previous works and difference between the two regimes (Reviewer #3).** The formulation we consider in the paper has already been studied in several works to better understand the behaviour of overparameterized neural networks and their optimization using SGD or simply gradient descent. However, the main difference between our work and previous studies is the use of a stepsize depending on the width of the neural network which leads to a different mean-field limit. In contrast to the one obtained previously, this limit has a diffusion term. This illustrates that SGD has a potential regularization effect. In future work, we plan to rigorously investigate this phenomenon by establishing generalization bounds for the two regimes and compare them.

**Comparison with other formulations (Reviewer #1).** In what follows we try to clarify the following remark of Reviewer 1: *"Depending on how one thinks about it, the learning rate in previous papers on infinite width SGD depends on the number of hidden units."*. Set $a_N = \sum_{k=1}^{N} F(w^{k,N}, x)$ and consider the two functionals $\mathscr{R}^N(w^{1:N}) = \int_{\mathsf{X} \times \mathsf{Y}} \ell(N^{-1}a_N, y)\mathrm{d}\pi(x,y)$, $\bar{\mathscr{R}}^N(w^{1:N}) = \int_{\mathsf{X} \times \mathsf{Y}} \ell(a_N, y)\mathrm{d}\pi(x,y)$. Let $(W_n^{1:N})_{n\in\mathbb{N}}$ be the SGD scheme associated with the minimization of $\mathscr{R}^N$ and $(\bar{W}_n^{1:N})_{n\in\mathbb{N}}$ the one associated with the minimization of $\bar{\mathscr{R}}^N$, defined by the recursions (1) $W_{n+1}^{k,N} = W_n^{k,N} - \gamma_1 N^{-1} \int_{\mathsf{X} \times \mathsf{Y}} \partial_1\ell(N^{-1}\sum_{k=1}^{N} F(W_n^{k,N}, x), y)\nabla F(W_n^{k,N}, x)\mathrm{d}\pi(x,y)$, (2) $\bar{W}_{n+1}^{k,N} = \bar{W}_n^{k,N} - \gamma_2 \int_{\mathsf{X} \times \mathsf{Y}} \partial_1\ell(\sum_{k=1}^{N} F(\bar{W}_n^{k,N}, x), y)\nabla F(\bar{W}_n^{k,N}, x)\mathrm{d}\pi(x,y)$. From these definitions, we get that no choice for the stepsizes $\gamma_1$ or $\gamma_2$ implies that $(W_n^{1:N})_{n\in\mathbb{N}} = (\bar{W}_n^{1:N})_{n\in\mathbb{N}}$ because of the different scalings in the loss function, *i.e.* $\sum_{k=1}^{N} F(W_n^{k,N}, x)$ is multiplied by $1/N$ in (1) and not in (2). As a result, there is no immediate link between the setting we consider with a stepsize which depends on the number of hidden units and the classical setting for SGD. However, in the specific case where $\partial_1\ell$ is positively homogeneous, further conclusions can be drawn. This is currently under investigation. We will mention this observation in the paper.

**Differentiability of the features (Reviewer #2).** In our paper, we assume some high order differentiability conditions for the feature function in order to derive our results. We suspect that our regularity assumptions can be relaxed but at the expense of significant technical complications. That is why we have decided to limit our theoretical study to the smooth case. Nevertheless, in our experimental analysis we used ReLU activation functions which are not differentiable to illustrate that our findings also hold empirically in the non-smooth setting.

**Comparison with other quantitative results (Reviewer #3).** We agree with reviewer 3 that previous works have established quantitative propagation of chaos results, see Mei et al. [2018, 2019]. However, we found our quantitative results to be of interest since the propagation of chaos results in [Mei et al., 2018, Theorem 3] hold in high probability for different criteria (Fortet-Mourier metric and risk evaluation). In that respect we believe that our results in the case where $\beta = 0$ complement the ones of Mei et al. [2018, 2019] and extend them in the case where $\beta \neq 0$. We will add this remark in the revised version of our paper to better acknowledge the results of Mei et al. [2018, 2019].

**Assumptions in the main document (Reviewer #4).** We tried to simplify A1 in the main document but the gain of space was negligeable and that is why we think that it is better to keep the general formulation we have for the moment. However, we acknowledge that a discussion on this set of assumptions is in order and we plan to add it in the revision of our manuscript.

[Meta-Review · NeurIPS 2020]

This paper studies SGD dynamics for two-layers neural networks by approximating it by a mean field diffusion focusing on a large width limit. In a nutshell the authors bound the distance between SGD iterates and that of the mean field dynamic. Using this theory the authors also study the effect of the scaling of the step size. All reviewers thought the paper was interesting, in particular the regime change result. Reviewer 2 had some concerns about the strictness of the assumptions which were mitigated based on the authors’ response. I concur with the positive reviews and recommend the paper to be accepted. I do want to note that the reviewers raised very valid points in their feedback including “extended discussion of how to interpret the learning rate in terms of alpha, beta, N, n” and “that generally the bounds produced in this article will grow exponentially with depth” raised by reviewer 1 and “It is unclear what insight emerges from this new analysis that was not already in previous work” from Reviewer 3. The authors should address these concerns in their final manuscript.